# *Drosophila* serotonin 2A receptor signaling coordinates central metabolic processes to modulate aging in response to nutrient choice

Yang Lyu[1], Kristina J Weaver[1], Humza A Shaukat[2], Marta L Plumoff[2], Maria Tjilos[2], Daniel EL Promislow[3,4], Scott D Pletcher[1]*

[1]Department of Molecular and Integrative Physiology and Geriatrics Center, Biomedical Sciences and Research Building, University of Michigan, Ann Arbor, United States; [2]College of Literature, Science, and the Arts, University of Michigan, Ann Arbor, United States; [3]Department of Lab Medicine & Pathology, University of Washington School of Medicine, Seattle, United States; [4]Department of Biology, University of Washington, Seattle, United States

**Abstract** It has been recognized for nearly a century that diet modulates aging. Despite early experiments suggesting that reduced caloric intake augmented lifespan, accumulating evidence indicates that other characteristics of the diet may be equally or more influential in modulating aging. We demonstrate that behavior, metabolism, and lifespan in *Drosophila* are affected by whether flies are provided a choice of different nutrients or a single, complete medium, largely independent of the amount of nutrients that are consumed. Meal choice elicits a rapid metabolic reprogramming that indicates a potentiation of TCA cycle and amino acid metabolism, which requires serotonin 2A receptor. Knockdown of *glutamate dehydrogenase*, a key TCA pathway component, abrogates the effect of dietary choice on lifespan. Our results reveal a mechanism of aging that applies in natural conditions, including our own, in which organisms continuously perceive and evaluate nutrient availability to promote fitness and well-being.

*For correspondence:
spletch@med.umich.edu

**Competing interests:** The authors declare that no competing interests exist.

## Introduction

For nearly all metazoans, many aspects of biology are continuously influenced by information about nutrient availability. Interest in the nutritional determinants of aging date back to the early 1900s, as experiments in fruit flies (*Chippindale et al., 1993*; *Chapman and Partridge, 1996*) and rodents (*McCay et al., 1935*; *Weindruch et al., 1986*; *Weindruch and Walford, 1982*) revealed that limiting total food intake or reducing diet quality, without malnutrition, increased lifespan. Dietary restriction (DR, also called caloric restriction) is now known to effectively increase the lifespan of nearly every species to which it has been applied. In mammals, increased lifespan in response to DR is accompanied by a broad-spectrum improvement in health during aging (*Weindruch and Walford, 1988*; *Longo and Finch, 2003*; *Barger et al., 2008*; *Contestabile, 2009*).

Research over the past century on the effects of DR has revealed how food characteristics influence healthy aging (*Fontana and Partridge, 2015*; *Tatar et al., 2014*). For many years, it was believed that the caloric content of the food and the amount of energy consumed by the animal were the driving forces behind the effects of restriction on lifespan (*Masoro, 2005*). More recently, however, conceptual advances such as the geometric framework for nutrition (*Lee et al., 2008*; *Solon-Biet et al., 2014*; *Skorupa et al., 2008*), together with comprehensive experiments that manipulate diet in model systems (*Skorupa et al., 2008*; *Bruce et al., 2013*; *Solon-Biet et al., 2014*;

**eLife digest** The foods we eat can affect our lifespan, but it is also possible that thinking about food may have effects on our health. Choosing what to eat is one of the main ways we think about food, and most animals, including the fruit fly *Drosophila melanogaster*, choose their foods. The effects of these choices can affect health via a chemical in the brain called serotonin. This chemical interacts with proteins called serotonin 2A receptors in the brain, which then likely primes the body to process nutrients.

To understand how serotonin affected the lifespan and health of fruit flies, Lyu et al. compared flies that were offered a single food to those that could choose between several foods. The flies that had a choice of foods lived shorter lives and produced more serotonin, but these effects were reversed when Lyu et al. limited the amount of a protein called glutamate dehydrogenase, which helps cells process nutrients.

These results suggest that choosing what we eat can impact lifespan, ageing and health. Human and fly brains share many similarities, but human brain chemistry is more complex, as is our experience of food. This work demonstrates that food choices can affect lifespan. More research into this phenomenon may shed further light onto how our thoughts and decision-making impact our health.

*Jensen et al., 2015*), have made it clear that dietary composition and availability of specific nutrients are often more relevant than calories in shaping life histories. For example, in homogenous nutritional environments where all nutrients are mixed together in a single food, animals fed a high carbohydrate–low protein diet generally live longer than siblings fed a calorically equivalent low carbohydrate–high protein diet (*Lee et al., 2008*; *Bruce et al., 2013*). Short-term experiments in heterogeneous nutritional environments, where animals are given a choice of two foods with different macronutrient ratios, reveal a behavioral preference for a diet that is high in carbohydrate and low in protein, suggesting that animals will adjust their dietary choices to optimize evolutionary fitness (*Lee et al., 2008*; *Jensen et al., 2015*; *Bunning et al., 2016*). The long-term effects of a complex nutritional environment and the accompanying dietary choices on physiology and healthy aging are unknown. Macronutrient intake will almost surely not be the only factor here because sensory perception of specific nutrients (*Ostojic et al., 2014*; *Waterson et al., 2014*), and even the presence of choice itself (*Avondo et al., 2013*; *Steenfeldt et al., 2019*), may be influential.

We have recently reported that how a meal is presented, or perhaps the way in which it is eaten, modulates aging in *Drosophila* (*Ro et al., 2016*). We found that flies were longer-lived when aged in an environment in which all dietary components were mixed together (termed a fixed diet) compared to one in which the two macronutrients, yeast and sugar, were available separately (termed a choice diet). When presented with a choice diet, flies consume an equivalent amount of protein and a higher carbohydrate:protein ratio than they do when fed a nutritionally equivalent fixed diet. Contrary to the results of most geometric nutritional studies (*Bruce et al., 2013*; *Lee et al., 2008*), however, their lifespan is shorter than those fed on high carbohydrate:protein fixed diets. This led us to speculate that the effects of dietary choice were derived, at least in part, from unknown effects of the complex nutritional environment.

Here we present a series of experimental results that indicate that dietary choice elicits significant changes in the physiology and lifespan of male *Drosophila melanogaster* through mechanisms that are distinct, at least in part, from nutrient consumption. Coincident changes in behavioral and health metrics suggest that flies exhibit distinct neural and metabolic states when presented with a choice diet than they do when presented with one that is nutritionally homogeneous. We report that neuronal expression of *serotonin receptor 2A* (*5-HT2A*) during the adult stage is required for many of the effects caused by dietary choice, including a 5-HT2A-dependent metabolic reprogramming that is consistent with a significant upregulation of the tricarboxylic acid (TCA) cycle and amino acid metabolism when nutrients were presented separately. Transgenic knockdown of a key enzyme, glutamate dehydrogenase (GDH), which is predicted to diminish the metabolic changes induced by a choice diet, abrogated its effects on lifespan. Future research may elucidate the downstream targets

through which neuronal 5-HT2A signals acts as a potent modulator of aging and physiology in an environment much like our own, where animals must constantly choose between different nutrients.

## Results

As with most model organisms, *D. melanogaster* reared in a conventional laboratory setting is normally provided with a homogenous, complete media that meets all of its nutritional needs. For flies, this medium is comprised predominantly of a source of protein and micronutrients (e.g., brewer's yeast) and a carbohydrate source (e.g., sucrose). In natural conditions, however, a complete food source is unlikely to be found, and animals must seek out and identify nutritionally complementary foods (*Csata et al., 2020*; *Mayntz et al., 2005*). Indeed, fruit flies are known to recognize and choose among different macronutrients based on palatability (*Weiss et al., 2011*; *Harris et al., 2015*) and physiological demand (*Ribeiro and Dickson, 2010*; *Vargas et al., 2010*; *Ro et al., 2016*; *Steck et al., 2018*). While studying the mechanisms underlying protein feeding, we observed significant modulation of survival depending on whether the animals were given a complete diet or were provided with a dietary choice by presenting the two macronutrient sources separately (*Figure 1A*, also see *Ro et al., 2016*). These effects were observed in both sexes (*Figure 1A*) and in different laboratory strains (*Supplementary file 1*). The magnitude of the choice effect was consistently greater in males (29–37% change in mean lifespan) than in females (5–15% change in mean lifespan, see *Supplementary file 1*). Similar effects were observed using two *D. melanogaster* lines that were recently collected from the wild: the lifespans of male strains were consistently and significantly reduced in response to dietary choice and there was variability in the female response (*Figure 1B*). The effects on lifespan are not due to some sort of acute toxicity because flies maintained on a choice diet for a short period (e.g., 2 weeks) exhibited general health metrics, such as intestinal integrity and climbing ability (*Figure 1C,D*), that were indistinguishable from siblings that were fed a fixed diet. Together these results indicate that the effects of dietary choice on lifespan are sex specific and are not an artifact of long-term laboratory culture.

Four lines of evidence indicate that the effect of a choice diet on lifespan is driven, at least in part, by mechanisms that are independent of increased sucrose consumption. First, increased sugar consumption alone is generally not associated with a large reduction in fly lifespan; even extremely sugar-rich diets (e.g., greater than 8× sugar intake) have only modest effects (*Dobson et al., 2017*), much smaller than we observed following dietary choice. New data that we obtained when we measured lifespan on fixed diets designed to approximate the nutritional composition that flies voluntarily constructed from their choice diet were consistent with this interpretation. We compared the lifespans of flies aged on a choice diet with those obtained from flies aged on one of two high-sugar, fixed diets (S30Y3 and S24Y3; i.e., 30% [w/v] sugar or 24% [w/v] sucrose mixed with 3% [w/v] yeast, respectively), among which flies consumed statistically indistinguishable amounts of both sugar and yeast (*Figure 1F*, represented as mass of nutrient consumed). The effect of dietary choice, relative to the standard SY10 diet, was much greater than that from homogeneous high-sugar diets (*Figure 1G*). Second, the effects of the choice diet on gross measures of fly physiology were distinctly different from those caused by increased sugar intake in the fixed diet. Flies exposed to a choice diet exhibited a significant reduction in triglyceride/protein ratio (*Figure 1H*) and triglyceride abundance (*Figure 1—figure supplement 1*; total weight was not affected; ), while increased sugar in the fixed diets had no such effect (*Figure 1H*, *Figure 1—figure supplement 1*). Third, we observed that simply changing the nature of the choice, by manipulating the concentration of the yeast component of the choice diet, was sufficient to modulate lifespan even though both cohorts of flies consumed the same amount of sugar and protein (*Figure 1I,J*, represented as mass of nutrient consumed). Moreover, the effect was opposite to that normally seen when similar manipulations were applied to a homogeneous diet (*Mair et al., 2005*); an increased concentration of the yeast component of choice extended lifespan (*Figure 1I*). Fourth, the transcription factor *dFoxo* is required both for transcriptional changes and for nutritional programming of lifespan caused by excess dietary sugar (*Dobson et al., 2017*), but it is dispensable for the lifespan effect of dietary choice (*Figure 1—figure supplement 2*). Together, these observations suggest that animals respond differently to macronutrients depending on how they are presented and that the effects of dietary choice on lifespan and physiology may be mediated by processes that are distinct from macronutrient intake.

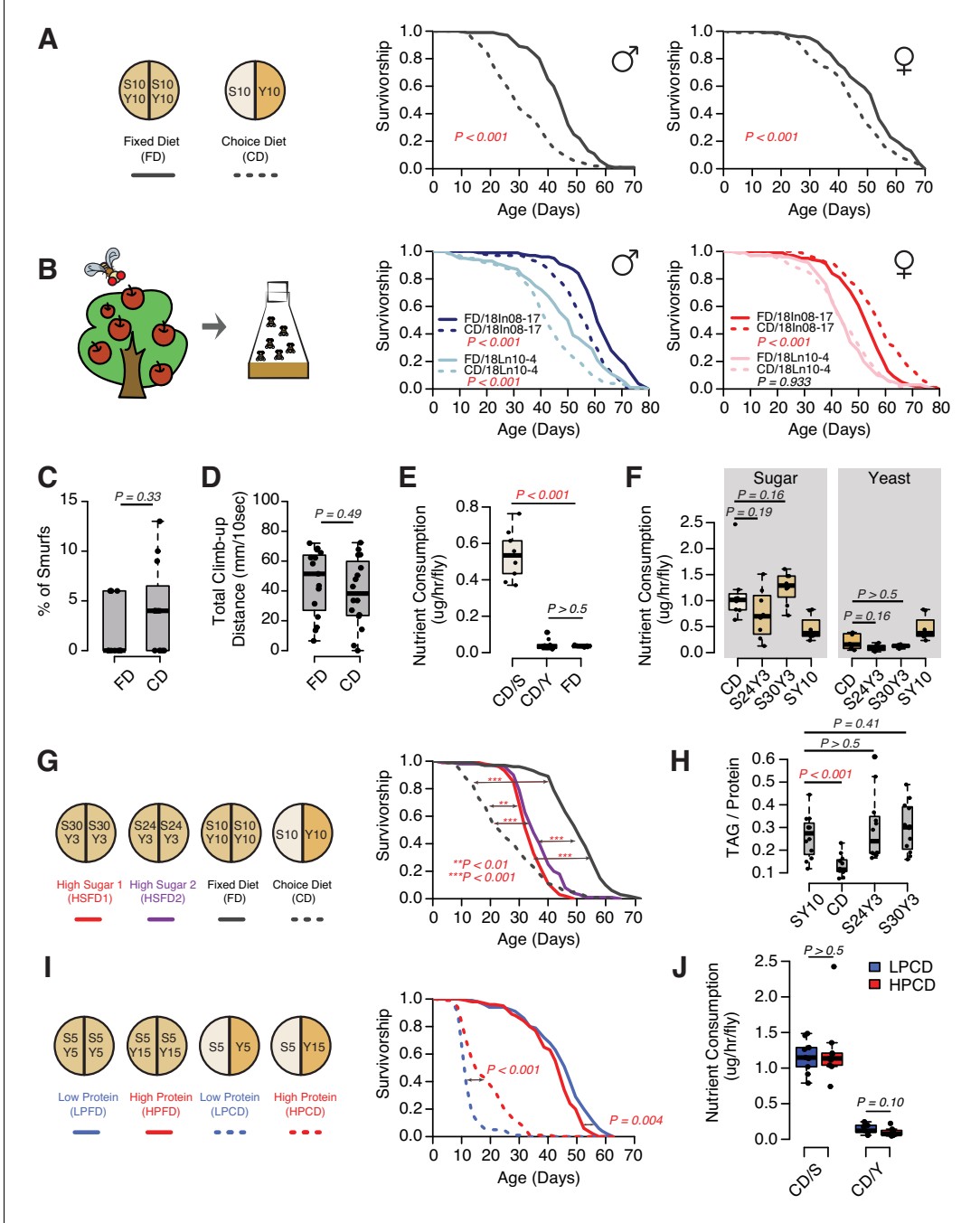

**Figure 1.** The presentation of nutrients significantly affects lifespan independent of the consumption of each nutrient in *Drosophila*. (A) Separating macronutrients in diet is sufficient to modulate aging. The left panel illustrates the two-well food containers for presenting either a fixed diet (FD, 10% [w/v] sugar and 10% [w/v] yeast mixed together) or a choice diet (CD, 10% [w/v] sugar and 10% [w/v] yeast presented on different sides). The right panel shows that both *Canton-S* males and females fed on the choice diet have shorter lifespans relative to their siblings that fed on the fixed diet (N = 104–112 per group). (B) Dietary effects are presented in two *Drosophila melanogaster* strains that were recently collected from the wild. Males from both strains exhibit a choice diet effect, whereas the responses of females are variable: one strain shows no lifespan difference between the two diets and the other one lives longer on a choice diet (N = 144–152 for each group). (C–D) Males fed on a choice diet exhibit the same intestinal integrity (C, N = 246–291 per diet) and climbing ability (D, N = 14 per diet) compared to fixed diet siblings. (E) Males on a choice diet consume more sugar and the same amount of yeast compared to their siblings that are fed on a fixed diet (N = 68–89 per diet). (F) Males on a choice diet consume a similar amount of sugar compared to their siblings on high-sugar fixed diets (S30Y3 and S24Y3; i.e., 30% [w/v] sugar or 24% [w/v] sucrose mixed with 3% [w/v] yeast, respectively; N = 65–82 per diet). (G–H) Dietary effects on lifespan (G, N = 144–149 for each group) and triglycerides/protein ratio (H, N = 60 for each group) differ between a choice diet and two high-sugar fixed diets. (I–J) Increased dietary protein in a choice diet is sufficient to promote longevity (I,

*Figure 1 continued on next page*

*Figure 1 continued*

N = 102–154 for each group) without affecting the macronutrient intake (J, N = 52–71). p-Values were obtained for lifespan using a log-rank test and for smurf assay, climbing ability, nutrient consumption, and TAG amounts using a Mann–Whitney U test. Boxplots in (C–F, H and J) represent the medians (the middle bar) and the first and the third quartiles (box boundaries) for each experimental group. Male flies were used in the experiments unless otherwise mentioned.

The online version of this article includes the following source data and figure supplement(s) for figure 1:

**Source data 1.** Effects of dietary choice on lifespan and behavior.

**Figure supplement 1.** Compared to a standard fixed diet SY10, either choice diet or high-sugar diet have an effect on weight.

**Figure supplement 1—source data 1.** Effects of dietary choice or dietary sugar on weight, TAG, and protein abundance.

**Figure supplement 2.** Two independent replicates support that the lifespan effect through dietary choice is largely independent of dFOXO (N = 126–150).

**Figure supplement 2—source data 1.** dFOXO lifespan data.

**Figure supplement 3.** Neither TOR activity (A) nor egg production (B) responds to a choice diet.

**Figure supplement 3—source data 1.** Egg production data.

Although protein intake was not different in flies fed different diets, it seemed possible that dietary choice improved protein absorption, which might be important considering the influential role of this macronutrient in reducing lifespan (*Fontana and Partridge, 2015*). Evidence suggests that this is not the case. Individual weight was not different among flies fed different diets (*Figure 1—figure supplement 1*), and total protein was lower in flies given a dietary choice than it was in their siblings fed a fixed diet, which is the opposite of what would be expected from the effect of dietary choice on lifespan. Furthermore, molecular and physiological markers that were strongly affected by protein availability, including target of rapamycin (TOR) activity (*Fontana and Partridge, 2015*) and female fecundity (*Jensen et al., 2015*), were not altered by dietary choice (*Figure 1—figure supplement 3*).

Previously, we hypothesized that neuronal mechanisms involved in nutrient evaluation might influence lifespan when macronutrients are presented separately (*Ro et al., 2016*). Our candidate screen of neurotransmitters and their receptors revealed that the presence of a *P*-element insertion in the promoter region of *serotonin receptor 2A* (*5-HT2A$^{PL00052}$*), which creates a putative loss of function mutation in this gene (*Nichols, 2007*), reduced protein preference in starved female flies and significantly extended lifespan under dietary choice (*Ro et al., 2016*). Loss of function in other serotonin receptors had no influence on protein preference (*Ro et al., 2016*), yet it remains to be tested whether they influence the aging processes in a choice environment. Here we focused on elucidating the mechanisms underlying the effects of the 5-HT2A receptor and expanded our analysis of its effects on dietary choice and male lifespan.

Indeed, we found that the *P*-element insertion in *5-HT2A* fully abrogated the male lifespan differences between fixed and choice dietary environments, largely by increasing lifespan when nutrients were presented individually (see *Figure 2A*). Mutant flies were modestly short-lived on the standard, fixed diet, which is likely due to the pleotropic effects of this receptor (*Gasque et al., 2013*; *Chakraborty et al., 2019*; *Howard et al., 2019*). We also investigated the influence of RNAi-based knockdown of *5-HT2A* expression using multiple RNAi lines. Because *5-HT2A* is highly expressed in the central nervous system (*Supplementary file 2*), we used the pan-neuronal driver *elav-GAL4* to achieve *5-HT2A* knockdown, and we observed that this manipulation abrogated the lifespan effect of dietary choice using one of the RNAi lines (*UAS-5-HT2A-RNAi$^{TRiP}$*) and strongly reduced the effect in the second (*UAS-5-HT2A-RNAi$^{KK}$*; *Figure 2B*). We also observed that the survivorship of adult flies fed with 200 μM pirenperone, an anxiolytic known to antagonize 5-HT2A receptor signaling in mammals, was unaffected by nutrient presentation, further supporting a role for this receptor and ruling out a necessity for loss of function during development (*Figure 2C*, *Figure 2—figure supplement 1* for the second replicate). Together, these results implicate neuronal expression of receptor *5-HT2A* in the adult male fly in mediating the effect of dietary choice on lifespan.

Although we concluded that nutrient intake was unlikely to be responsible for the difference in lifespan in wild-type flies, it remained formally possible that feeding differences between control and *5-HT2A* mutant flies were responsible for the dissimilar response to dietary choice. Our previous work demonstrated that 5-HT2A influenced protein preference after starvation (*Ro et al., 2016*), and we asked whether it influenced nutrient consumption in fed flies. Surprisingly, we found that, in the

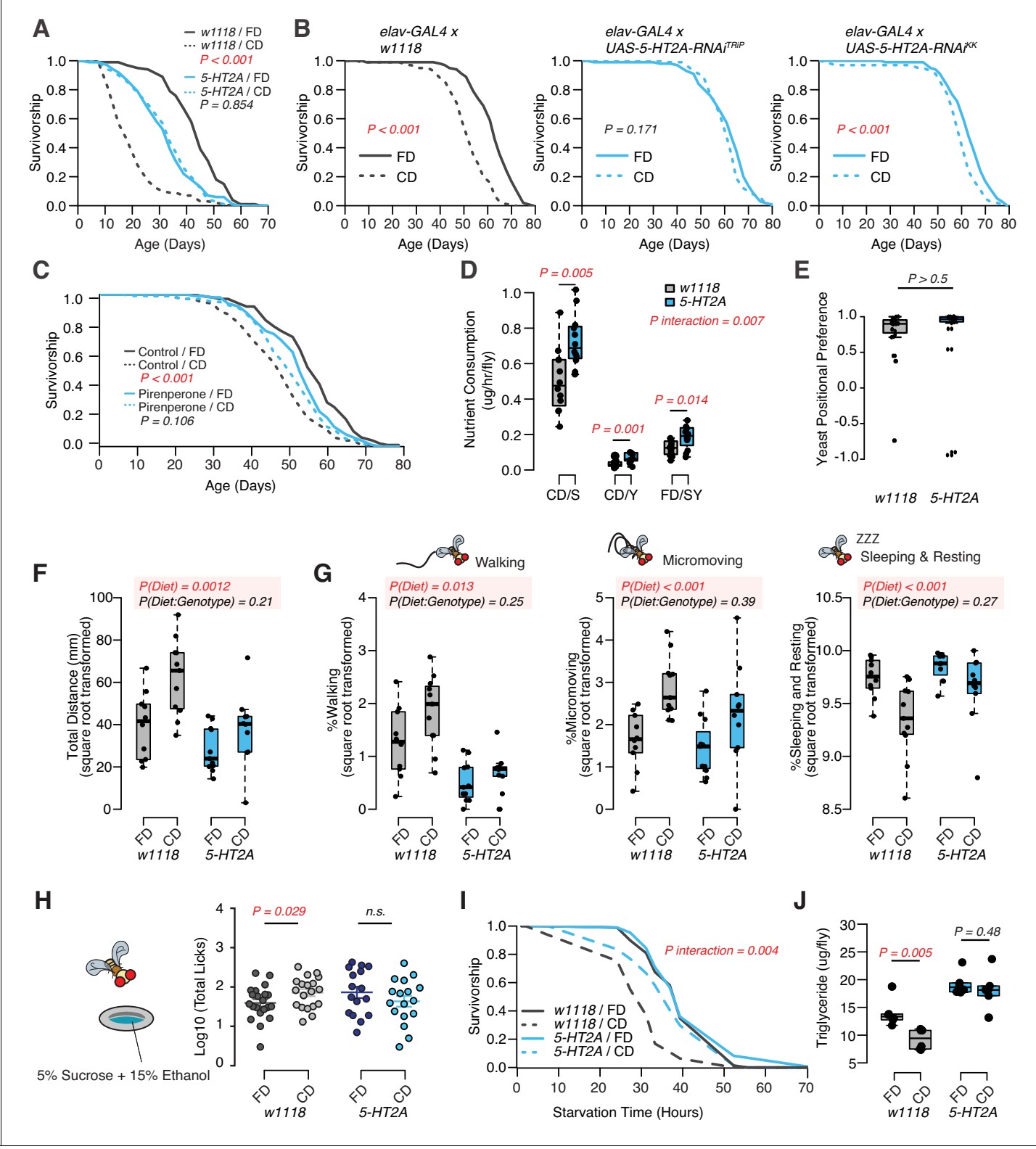

**Figure 2.** Dietary choice influences lifespan, behavior, and physiology through the serotonin receptor 2A (5-HT2A). (**A**) *P*-element insertion in *5-HT2A* abrogates the differences in male lifespan caused by dietary choice (N = 145–148 per group). (**B**) Pan-neuronal knockdown of *5-HT2A* expression eliminates (with *UAS-5-HT2A-RNAi^TRiP^*) or reduces (with *UAS-5-HT2A-RNAi^KK^*) the choice effect compared to control flies (N = 145–150 per group). (**C**) Pharmacological treatment of *Canton-S* males with a serotonin receptor 5HT₂-antagonist, pirenperone, during adulthood effectively abrogates the

*Figure 2 continued on next page*

*Figure 2 continued*

lifespan differences caused by dietary choice (N = 145–153 per treatment). (D) *5-HT2A* males exhibit similar feeding preferences on the choice diet, but consume more nutrients than *w^1118* animals on both diets (N = 97–115 for each group). (E) Both *w^1118* and *5-HT2A* males spent a majority of their time on yeast food in a choice environment (PI = ($T_{yeast}$ − $T_{sugar}$)/($T_{yeast}$ + $T_{sugar}$); N = 24 for each group). (F–G) Males in a choice environment exhibit increased activity: they traveled farther (F), demonstrated more walking and micromovements, and spent less time sleeping and resting (G) when they are exposed to choice diet, and such effects are independent of 5-HT2A (N = 10 or 11 per group.) (H) A schematic illustrating the ethanol feeding assay. Males were fed on choice diet or fixed diet for 10 days before being placed in chambers where they could feed from wells containing food solutions with 15% ethanol (refer to the Materials and methods for more details). *w^1118* flies previously fed on choice diet show significantly more licks on ethanol-containing food than sibling flies fed on fixed diet. *5-HT2A* mutants show the same trend independent of the diet they previously fed on (N = 16–21 for each treatment). (I–J) One to 2 week's feeding on choice diet decreases starvation resistance (I, N = 72–88 per group) and TAG amounts (J, N = 48 per group), and both effects require 5-HT2A. p-Values were obtained for lifespan and starvation resistance using a log-rank test, for nutrient consumption and activity using a two-way ANOVA analysis, for positional preference using a Mann–Whitney U test, and for ethanol feeding using a one-tail Student's t-test. Boxplots in (D, E, F, G, H, and J) represent the medians (the middle bar) and the first and the third quartiles (box boundaries) for each experimental group.

The online version of this article includes the following source data and figure supplement(s) for figure 2:

**Source data 1.** Requirement of 5-HT2A in the dietary choice effects.
**Figure supplement 1.** Second replicate to validate the role of pirenperone in abrogating the lifespan effect of dietary choice.
**Figure supplement 1—source data 1.** Lifespan data from pirenperone adminstration.

choice environment, *5-HT2A* mutant flies exhibited feeding characteristics similar to control animals: each consumed significantly more sucrose than yeast when given a dietary choice (*Figure 2D*), despite spending the majority of their time physically located on the yeast food (*Figure 2E*). Unlike control animals, however, the behavioral differences in *5-HT2A* mutants were not coupled to changes in lifespan.

DR, and the extended lifespan that accompanies it, are associated with increased locomotion behavior in several species (*Weindruch et al., 1986*; *Weed et al., 1997*; *Chen, 2005*), presumably reflecting a motivational state of hunger. By video tracking control and mutant flies in each of the dietary environments (using our DTrack system, see Materials and methods), we found that control flies were generally more active when given a dietary choice: they traveled significantly farther in a 6 hr window when compared to siblings housed on a fixed diet. *5-HT2A* mutant flies responded similarly (*Figure 2F*, two-way ANOVA on the square root transformed values, $p_{Diet*Genotype} > 0.05$). Behavioral classification of the movements during this time (*Corrales-Carvajal et al., 2016*) revealed that diet-dependent differences were mirrored in both the time spent walking (speed > 2 mm/s), in executing micromovements (e.g., grooming or when 0.2 mm/s < movement speed ≤ 2 mm/s), and in the time spent sleeping or resting (speed ≤ 0.2 mm/s) (*Figure 2G*). As with total distance moved, *5-HT2A* mutants exhibited similar responses as control animals in all of these measures (*Figure 2G*, two-way ANOVA on the square root transformed values, $p_{Diet * Genotype} > 0.05$). From these data we concluded that diet-dependent changes in locomotion are not causal to changes in lifespan.

Increased locomotor behavior has been linked with stress in mice (*Sestakova et al., 2013*) and perhaps in flies (*Mohammad et al., 2016*), leading us to examine whether other stress-related phenotypes were changed in flies fed a choice diet (*Strekalova et al., 2005*; *Linford et al., 2015*; *Yang et al., 2015*). First, we measured alcohol consumption because ethanol intake is known to be potentiated by conditions of stress or reward depletion (*Devineni and Heberlein, 2009*; *Shohat-Ophir et al., 2012*). We found that when flies were provided a dietary choice for 10 days prior to being exposed to sucrose food containing 15% (v/v) ethanol, they interacted with the ethanol food significantly more than did flies previously fed a homogenous fixed diet (*Figure 2H*, see Materials and methods for more details). *5-HT2A* mutant flies did not exhibit a difference in ethanol food interactions based on previous diet (*Figure 2H*). Second, we measured triglyceride (TAG) abundance (*Figures 1H* and *2J*), which has been postulated to be a physiological indicator of stress (*Starzec et al., 1981*; *Starzec and Berger, 1986*). We found that flies given a dietary choice had reduced TAG and lived shorter following starvation on 2% agar than did flies fed a homogenous food (*Figure 2I*). Both phenotypes were 5-HT2A dependent (*Figure 2I ,J*).

Our results indicated that 5-HT2A is not involved in determining how animals interact with the choice environment, yet the requirement of this neuronal signaling in lifespan (*Figure 2B*) and physiological traits (*Figure 2I,J*) suggested that it may play a role in mediating internal nutrient

homeostasis when flies are presented with a dietary choice. We therefore sought to identify changes in metabolite abundance using targeted metabolomics. To delineate an appropriate window of time to study, we completed a demographic analysis of the effect of dietary choice on age-specific mortality. Mortality rates of male flies whose nutrients were presented separately following eclosion exhibited measurable and high mortality rates by day 10, while similar levels of mortality in flies exposed to a homogenous food did not appear until day 40 (*Figure 3A*, compare black and orange solid lines). To determine how quickly the effects of nutrient presentation manifest, we performed a switch experiment in which flies fed a homogenous medium since eclosion were given a dietary choice beginning on day 20. We observed that the mortality rate of the switched cohort rose within 48 hr after choice began, and after roughly 5 days, it reached the level of the cohort whose nutrients had been separate since eclosion. We also performed the opposite experiment in which nutrients were combined into a fixed diet at day 20. Flies in this cohort exhibited a rapid decrease in mortality, again within 48 hr. Mortality rates in this cohort reached the level observed for flies that had been fed a homogenous food since eclosion, but this process took roughly 10 days (*Figure 3B*), which was longer than the reverse (*Figure 3A*). Cohorts that comprised *5-HT2A* mutant animals exhibited similar patterns of mortality regardless of nutrient presentation, and these patterns were also unaffected by dietary switch (*Figure 3C,D*).

We reasoned that metabolic changes that are causally related to aging would be identified by focusing on those that were abrogated by loss of 5-HT2A. Others, perhaps those triggered by differences in feeding behaviors or food intake (and independence of the lifespan effect), would persist in the two strains. We also chose to sample flies 48 hr after a dietary switch because this period was sufficient to significantly affect mortality rates (see *Figure 3A and B*) and because we felt that a comparison of switched to unswitched cohorts at this time would help further isolate changes that were causal to lifespan (*Figure 3E*). Finally, although neural expression of *5-HT2A* was required for the lifespan effects of dietary choice (*Figure 2B* above), it is also putatively expressed in salivary glands, in digestive tissues such as the crop, and in reproductive organs (*Supplementary file 2*). We therefore collected heads and bodies separately to determine whether the metabolomic effects of the two nutritional environments were enriched in the brain or whether they also manifested in peripheral tissues.

Using principal component analysis (PCA), we identified three principal components (PCs) in each of the tissue samples by which metabolome profiles effectively separated genotypic effects from those caused by nutrient presentation (additional PCA are presented in *Figure 3—figure supplement 1* and *Figure 3—figure supplement 2*). Genotype effects on the metabolome were demonstrated by PC2 and PC3 in the head and body data, respectively (*Figure 3F,G*). A second PC in each data set identified diet history (PC3 and PC2 in heads and bodies, respectively; *Figure 3F,G*). These combinations of metabolites provided separation based on how nutrients were presented to the flies for the 20 days prior to the dietary switch, and they were largely unaffected by the switch itself. Separation was also observed in *5-HT2A* mutant flies, suggesting that the metabolites that load strongly on these axes exhibit slow, persistent changes in response to differences in nutrient presentation that are not associated with changes in mortality. Finally, a single PC in both heads and bodies (represented by PC6 in both data sets) distinguished acute responses to dietary switch in both tissues that were abrogated in *5-HT2A* mutant animals (*Figure 3H,I*). These PCs identified specific metabolites in either heads or bodies that were correlated with changes in mortality and that shared a common 5-HT2A dependence.

The PCA results suggested that nutrient presentation influenced two processes: (A) a relatively stable 5-HT2A-independent metabolic network that responded slowly to dietary environment and that accounted for a significant amount of metabolome variation (10.2–11.2%) and (B) a dynamic 5-HT2A-dependent metabolic network that accounted for a smaller component of variation (2.0–2.2%) that responded quickly to dietary choice and that was associated with mortality. Using generalized linear models, we identified individual metabolites that exhibited statistically significant changes consistent with membership in each of these two groups (a summary of the analysis of a third group that distinguished genotype effects is presented in *Figure 3—figure supplement 3*; statistics for each metabolite are presented in *Supplementary file 3*). Criteria for inclusion in Group A consisted of a statistically significant effect of diet ($p_{Diet}<0.05$) and a non-significant diet × genotype interaction ($p_{Diet*Genotype}>0.05$). Of the 103 and 125 metabolites that were detected in each tissue sample, respectively, we found that 45 (heads) and 46 (bodies) satisfied these criteria (*Figure 4A*). Among

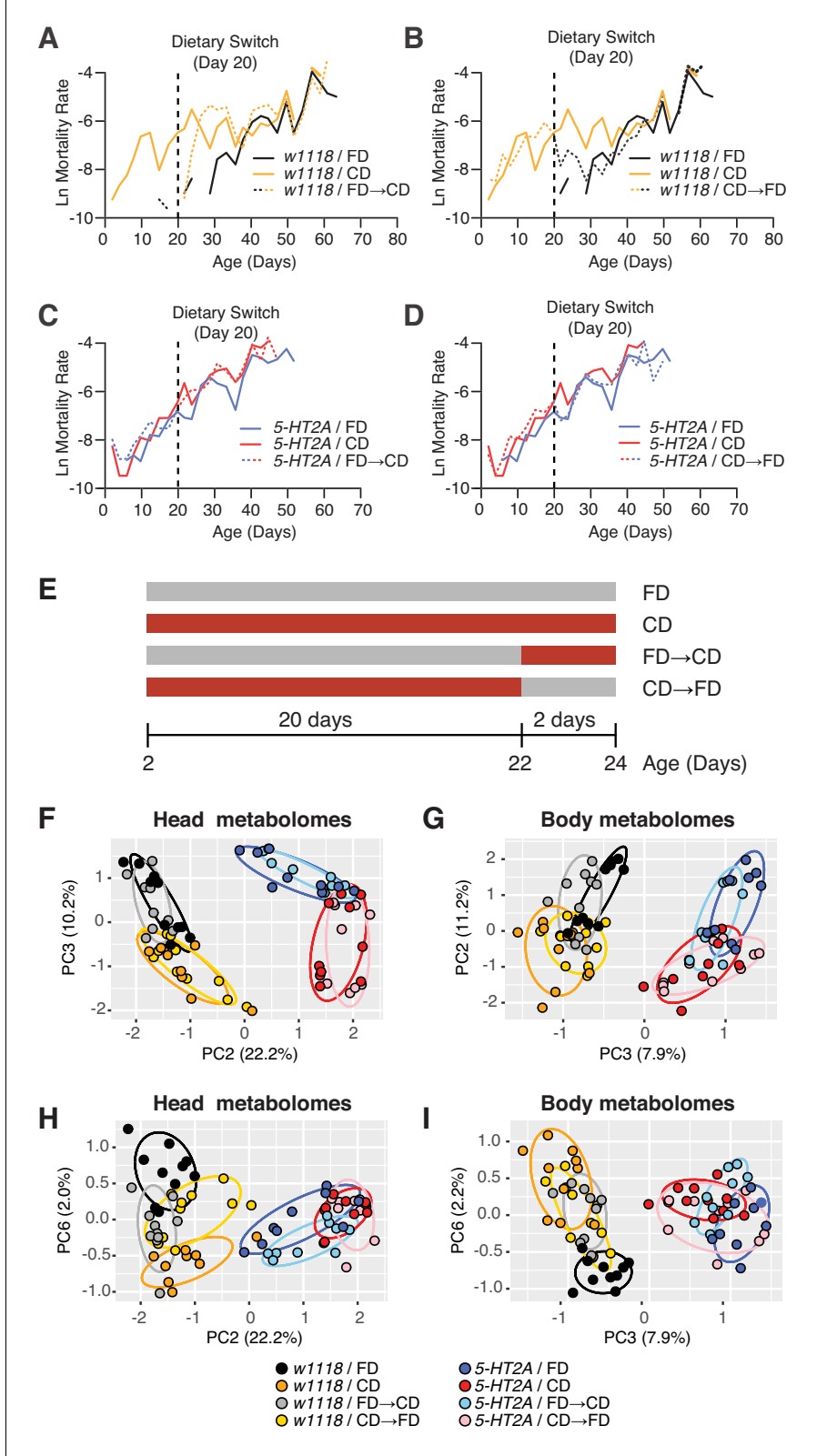

**Figure 3.** Temporal effects of dietary choice on mortality and metabolomes. (**A–D**) 5-HT2A mediates mortality responses to the introduction or removal of a choice diet. Mortality rate is plotted on the natural log scale (N = 225–282). Male flies are either fed on a choice diet (CD) or a fixed diet (FD) at the onset of adulthood. The dietary switch regime, either switched from FD to CD or switched from CD to FD, was applied at 20 days of

*Figure 3 continued on next page*

*Figure 3 continued*

adulthood, where the mortality rate of *w1118* males fed on CD is significantly higher than *w1118* males fed on FD. (**A**) *w*$^{1118}$ flies that previously fed on FD showed a rapid increase in mortality within 48 hr when switched to CD at 20 days of adulthood, compared to those of the same age that continually fed on FD. About 5 days after the switch, the switched cohorts were indistinguishable from those of the same-age flies maintained on a CD through adulthood. (**B**) Reciprocal switches from CD to FD resulted in immediate reduction in mortality rate within 48 hr. At 30 days of adulthood, the mortality rate of the switched cohorts was virtually the same as flies that fed on a CD throughout the adulthood. (**C–D**) *5-HT2A* flies from different dietary groups did not show differences in mortality rate, suggesting that the mortality effect is mediated through serotonin signaling. (**E**) Dietary switch paradigm illustrated at the different time points for metabolomics samples. (**F–I**) Principal component analysis (PCA) isolated the metabolome component that encodes the memory of previous diet and the component that represents the response to new diet. PCA plot shows the distribution of samples for each treatment in the head metabolomes and in body metabolomes. In the head (**F–H**), metabolites weighted heavily in PC3 (in the top panel), PC6 (in the bottom panel), and PC2 (in both panels) distinguish the memory of previous diet, the response to new diet, and the genotype, respectively. In the body (**G–I**), the effect of previous diet, the effect of new diet, and the effect of genotype were distinguished by the metabolites heavily weighted in PC2, PC6, and PC3, respectively (N = 8–10 samples for each treatment, and 30–50 heads or bodies were used for each sample).

The online version of this article includes the following source data and figure supplement(s) for figure 3:

**Source data 1.** Mortality data.

**Figure supplement 1.** PCA analyses in head metabolomes, including the proportion of variance of each PC and the PCA plots for the first six PCs.

**Figure supplement 2.** PCA analyses in body metabolomes, including the proportion of variance of each PC and the PCA plots for the first six PCs.

**Figure supplement 3.** Clustering of the abundance of each metabolite that differed between genotypes in head or body metabolomes.

---

those metabolites, 13 (heads) and 21 (bodies) exhibited an increase in abundance when nutrients were presented independently, while abundances of the remaining metabolites were reduced. Although our sample of measured metabolites was small and precluded a comprehensive pathway analysis (see Materials and methods), MetaboAnalystR (*Pang et al., 2020*) applied to the full list of significant metabolites revealed several candidate pathways (i.e., those with more than three significantly affected metabolites in each), nearly all involved in amino acid metabolism, which may be specifically affected by dietary choice (*Figure 4C*, see Materials and methods). Over 40% of the metabolites (N = 19 of 45 or 46 metabolites in the head or body, respectively) were modulated in both samples (starred in *Figure 4A*). While 75% of these (N = 14) were modulated in the same direction, there remained nearly a quarter of them that exhibited opposite changes in abundance.

Changes in Group B, which represented the 5-HT2A-dependent metabolome, were fewer in number but more consistent across tissue samples. We identified 18 of 103 total metabolites in the head and 19 of 125 total metabolites in the body that were present in this group (*Figure 4B*, ANOVA, $p_{Diet*Genotype} \leq 0.05$). In this case, six of seven of the metabolites that were present in both groups (starred in *Figure 4B*) were modulated in the same direction. Furthermore, MetaboAnalystR identified related pathways, involving the synthesis of aminoacyl-tRNAs or specific amino acids, which were found either in both tissues or only in the body (*Figure 4D*). Among them, 'alanine, aspartate and glutamate metabolism' was the only category of interest in both tissues, specifically in a 5-HT2A-dependent manner. A closer look into its metabolic map (KEGG ID: map00250) revealed that candidate metabolites are either members of the TCA cycle or the direct precursors of TCA metabolites, which indicated the modulation of this metabolic process in response to dietary choice.

A detailed examination of 5-HT2A-dependent metabolites, and the relationships among them, indicated significant upregulation of the TCA cycle and amino acid metabolism when nutrients were presented separately (*Figure 5A*). The TCA cycle is a series of biochemical reactions that generates energy (ATP) through the oxidation of acetyl-CoA derived from different nutrients, while the same reactions also provide precursors for amino acids (*Figure 5B*). Specifically, we found that oxaloacetate abundance was increased in both tissue samples in a 5-HT2A-dependent manner, while α-ketoglutarate and fumarate were upregulated only in heads and bodies, respectively. Further analysis revealed consistency among the abundances of the amino acid precursors of these intermediates, which were also increased in their corresponding tissues (*Figure 5A,B*). Asparagine, the immediate

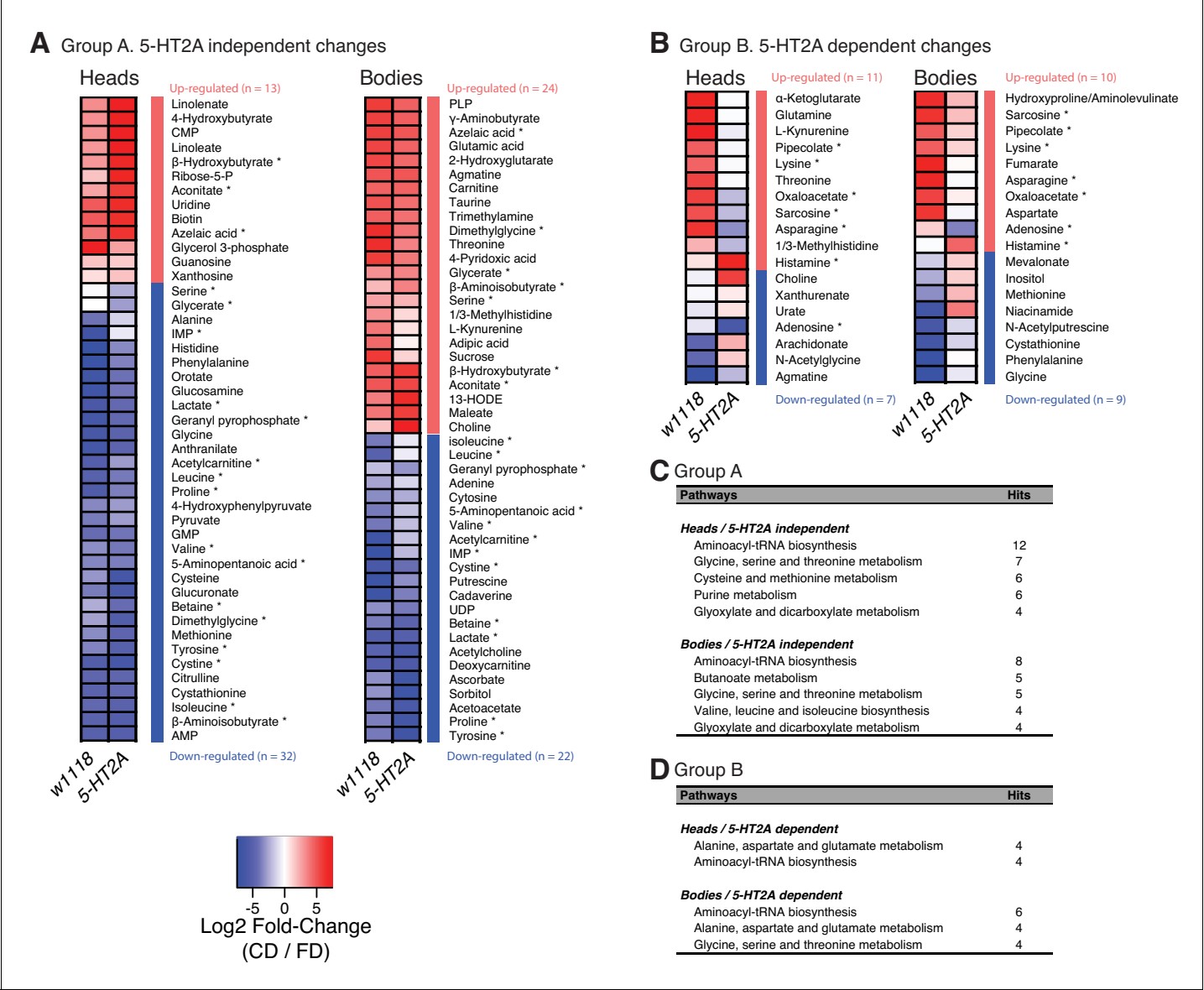

**Figure 4.** Nutrient presentation mediates metabolic changes in both head and body. (A–B) Generalized linear model separates the metabolites that respond to dietary choice in a 5-HT2A-independent (A) or -dependent (B) manner. Up- or down-regulation of metabolites upon choice are represented by red and blue, respectively. Metabolites that were altered in both heads and bodies are indicated by asterisks. (C–D) Metabolic pathways that are influenced by choice diet are revealed by MetaboAnalystR. Pathways with a number of hits > 3 are shown.

precursor for oxaloacetate, was increased in both heads and bodies; glutamine, the major precursor for α-ketoglutarate, was increased only in heads; and aspartate, the precursor of fumarate, was increased only in bodies. Although we were unable to measure abundance of acetyl-CoA directly, we observed that the abundance of lysine, its precursor, was increased in both tissue samples.

Inspired by concurrent changes in TCA intermediates and their precursors, we investigated the direct roles of biochemical reactions that produce the TCA metabolites in the modulation of aging in response to dietary choice. We targeted the enzyme GDH, which converts glutamate to α-keto-glutarate, for RNAi-mediated knockdown. Our metabolomic data revealed significantly increased abundance of α-ketoglutarate in the heads of animals given a dietary choice, and α-ketoglutarate has been shown to modulate longevity in *Caenorhabditis elegans* (*Chin et al., 2014*). Knocking-down GDH abundance in adult flies using the RU486-dependent GeneSwitch system (*Roman et al., 2001*) (*tub-GS-GAL4*) coupled with *UAS-GDH-RNAi* blocked the lifespan differences between two

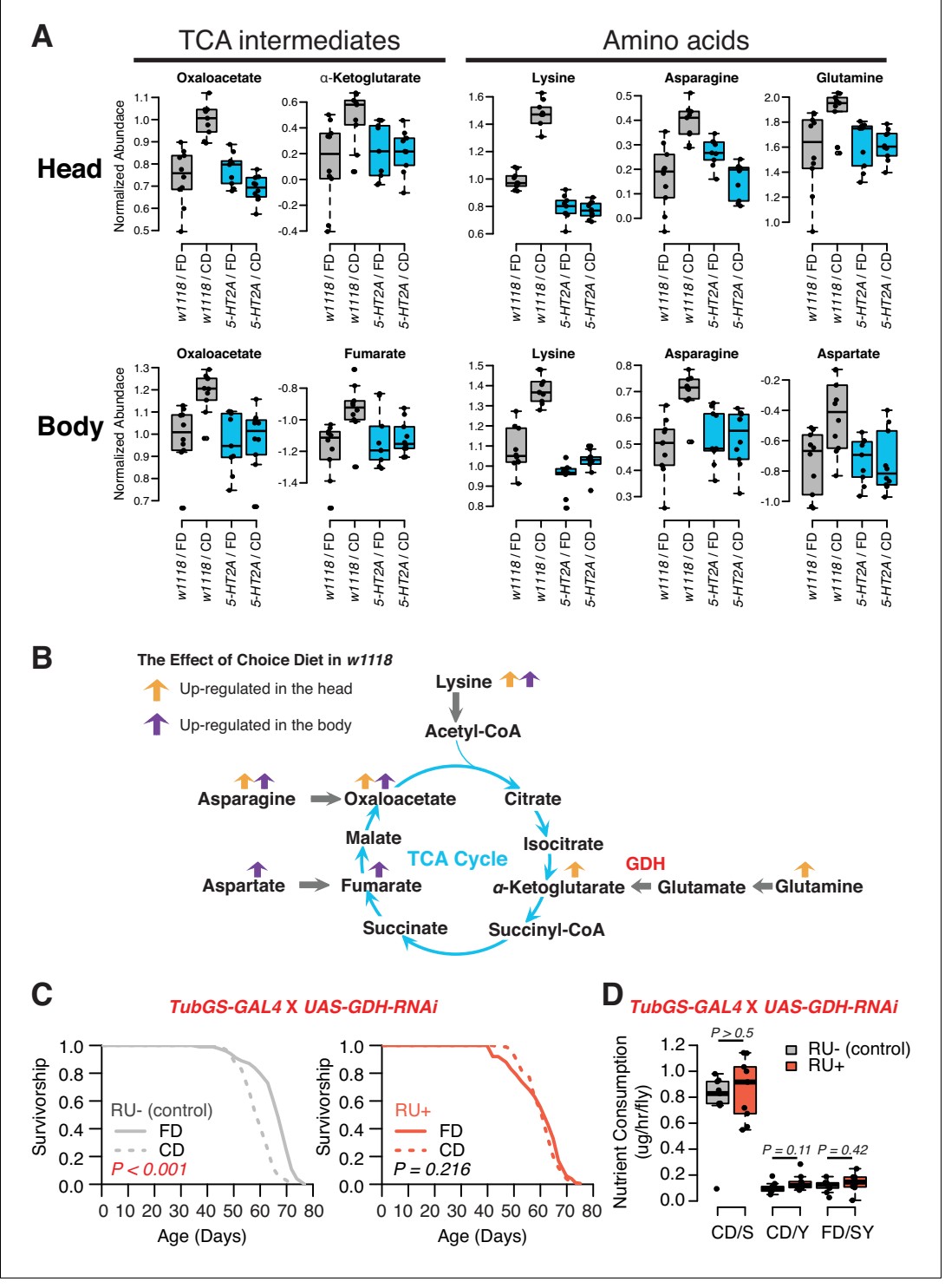

**Figure 5.** Nutrient presentation modulates central metabolic pathways through serotonin signaling. (**A**) Boxplots show the TCA intermediates and amino acids that are modulated by choice diet in a 5-HT2A-dependent manner (two-way ANOVA, p<0.05 for the interaction between diet and genotype). (**B**) The metabolites of the TCA cycle. Orange and purple arrows indicate the metabolites that are regulated in head and body when the flies are given choice diet conditions, respectively. (**C**) Ubiquitous knockdown of *glutamine dehydrogenase* (*GDH*) using the GeneSwitch system fully abrogates the lifespan differences between the two diets when RU-486 was administrated in the food (N = 144–153 per group; p-values are calculated using a log-rank test). (**D**) Knockdown of *GDH* has no

*Figure 5 continued on next page*

*Figure 5 continued*

effects in nutrient consumption on either diet (N = 71–94 per group, p-Values are calculated using a Mann–Whitney U test).

The online version of this article includes the following source data and figure supplement(s) for figure 5:

**Source data 1.** Lifespan and feeding data from *GDH* knock-down.
**Figure supplement 1.** RU has no effects in the lifespan and feeding.
**Figure supplement 1—source data 1.** Lifespan and feeding data from RU administration.
**Figure supplement 2.** Knockdown *GDH* has no effects on the protein amount of flies.
**Figure supplement 2—source data 1.** Protein data from *GDH* knock-down.

dietary environments (*Figure 5C*). This was not caused by the transcriptional inducer, RU486, because control animals (*tub-GS-GAL4* coupled with *UAS-GFP*) responded as expected to dietary choice when RU486 was presented (*Figure 5—figure supplement 1*). *GDH* knockdown had no effect on nutrient consumption (*Figure 5D*) or total protein abundance in the fly (*Figure 5—figure supplement 2*). Which tissues/cells are critical for the diet-specific lifespan effects induced by *GDH* knockdown, and whether such influences are due to changes in α-ketoglutarate per se or to alterations in the metabolic network more broadly (*Hoffman et al., 2017*) remain to be determined.

## Discussion

Here we studied the effects of meal presentation on aging and physiology in *Drosophila*. By separating the two macronutrients of the standard *Drosophila* laboratory diet (i.e., brewer's yeast and sucrose) and allowing the animals to behaviorally structure their own diet, we found that lifespan was significantly shortened in both male and female flies. In addition, changes in activity, stress resistance, lipid metabolism, and critical metabolic processes were also elicited by dietary choice. Flies carrying a loss-of-function mutation in *serotonin receptor 5A* (*5-HT2A*), with RNAi-mediated neuronal knockdown of *5-HT2A* expression, or fed the putative receptor antagonist pirenperone, exhibited no differences in lifespan between choice and fixed diets, indicating that serotonin signaling through this receptor plays an essential role in the effect of nutrient presentation on aging.

Flies voluntarily consumed more sugar than yeast when presented with a choice diet. Although it is difficult to definitively rule out differences in consumption as a cause for the phenotypes we observe, several lines of evidence indicate that is not the major factor influencing lifespan: (1) the effect of dietary choice on lifespan is significantly larger than that caused by even the most extreme carbohydrate-rich fixed diets (as shown in both *Figure 1G* and previous studies from *Skorupa et al., 2008*; *Dobson et al., 2017*); (2) the choice diet induced physiological effects (e.g., reduced triglyceride abundance) opposite of those associated with a high-sugar fixed diet (e.g., increased or unchanged triglyceride abundance depending on the genetic background, see *Skorupa et al., 2008*); (3) the insulin-responsive transcription factor, *dFoxo*, which is an essential effector of dietary sugar on aging (*Dobson et al., 2017*), is not required for the effect of dietary choice on lifespan; (4) manipulating the nature of the choice is sufficient to influence lifespan without altering the amount of nutrients eaten in a choice paradigm; and (5) flies with loss of *5-HT2A* exhibited the same diet-dependent changes in feeding as did control animals, yet they experienced no diet-induced changes in lifespan (*Figure 2A,D*).

We found that dietary choice influenced lifespan in both male and female laboratory flies and that 5-HT2A is required for this effect in both sexes (see female data in *Ro et al., 2016*), suggesting the mechanisms are at least partially shared between them. However, sex-specific phenomena were also observed: lifespan was significantly shortened in male flies for all strains tested (from 11 to 37% in laboratory and wild strains), whereas the effects on lifespan were variable in recently wild-caught females (an average of 3% reduction) relative to a fixed diet. This is, perhaps, not surprising given the effects of different macronutrients among different strains (*Jin et al., 2020*) and sexes (*Hudry et al., 2019*; *Camus et al., 2019*; *Jensen et al., 2015*). Future studies that focus on dissecting the role of individual macronutrients on choice effects or on investigating the involvement of sex-specific nutrient sensing proteins (see *Hudry et al., 2019*), perhaps with a focus on population genetics of natural populations, will be useful for understanding the nature of sexual dimorphism on lifespan and physiology.

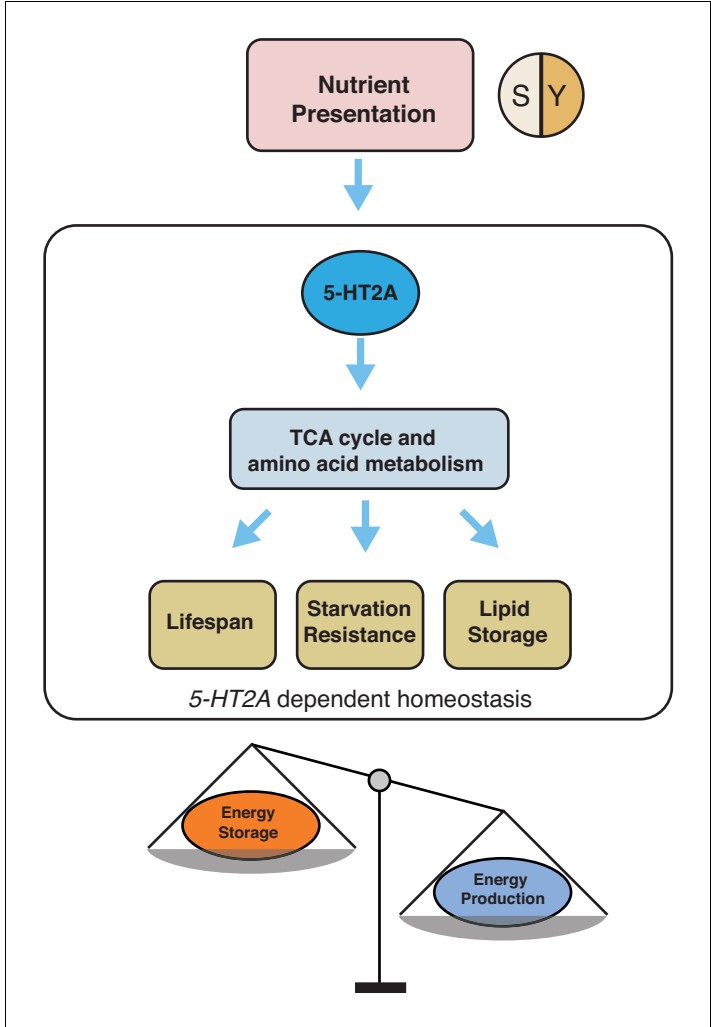

**Figure 6.** Our working model showing how 5-HT2A regulates energy homeostasis to influence aging and physiology in response to nutrient presentation.

We found that dietary choice led to a rapid and reversible change in age-specific mortality only during a defined period of life. Dietary choice significantly increased mortality within 48 hr of exposure, and this effect was observed only up until middle age (roughly 40 days in our experiments) after which time mortality rates in cohorts provided a fixed or choice diet were indistinguishable. This was true regardless of whether the flies had been presented with a dietary choice starting at eclosion or starting at 20 days of age, and these effects required serotonin receptor 5-HT2A. The effects of dietary choice on demographic aging, therefore, do not appear to accumulate, and they are simply absent in older flies, perhaps because of an age-dependent reduction in serotonin signaling, a lack of consequences of such signaling in older animals, or the manifestation of demographic heterogeneity (*Vaupel and Yashin, 1985*). These mortality patterns are distinct from those observed following other types of dietary or environmental perturbations. DR in flies is typically carried out by diluting both the yeast and carbohydrate components of a complete, fixed diet (*Chapman and Partridge, 1996*). DR rapidly influences mortality dynamics, and it is reversible at early ages. Unlike dietary choice, however, DR influences demographic aging across the whole of adulthood (*Mair et al., 2003*). The effect of different temperatures on mortality is cumulative and permanent (*Mair et al., 2003*; *Carvalho et al., 2017*).

The neuronal response to dietary choice reshapes metabolic networks that are associated with energy homeostasis in peripheral tissues (see our model in *Figure 6*). Some of these changes are mediated by 5-HT2A, which is widely expressed in the central nervous system (*Gnerer et al., 2015*) and in the thoracicoabdominal ganglion (*Supplementary file 2*). Our metabolomic data revealed that TCA cycle components, as well as key components of amino acid metabolism, are strongly affected by food choice and that such changes require 5-HT2A. There is evidence that at least one such metabolite, α-ketoglutarate (αKG), is directly involved in the effect of dietary choice on lifespan. αKG was more abundant when flies were presented with a choice diet, and inhibition of GDH, which is involved in αKG synthesis, abrogated the difference in lifespan between choice and fixed diets (see also *Talbert et al., 2015*). In *C. elegans*, it has been reported that αKG modulates longevity by inhibiting TOR signaling (*Chin et al., 2014*) and that oxaloacetate increases lifespan through an AMPK/FOXO-dependent pathway (*Williams et al., 2009*). The effect of αKG on lifespan has recently been independently documented in flies (*Su et al., 2019*), and, more recently, in mice (*Asadi Shahmirzadi et al., 2020*), suggesting a conserved role for αKG in modulation of aging. Published data in *Drosophila* suggest that supplementation of αKG to a standard laboratory diet increased transcript abundance of AMPK and FOXO and reduces that of TOR (*Su et al., 2019*). However, our data indicate that neither FOXO (*Figure 1—figure supplement 2*) nor TOR (*Figure 1—figure supplement 3*) signaling is involved in the effects of dietary choice on lifespan, suggesting that αKG may recruit different mechanisms in this environment. Perhaps instead of energy sensing pathways, metabolic enzymes that control the branch point of crucial reactions (such as GDH) are more relevant in the context of a choice environment (*Smith et al., 2019*).

We propose the idea that the effects of dietary choice may result from metabolic changes stimulated by an alteration in metabolic state induced by the need to continuously evaluate macronutrient availability against physiological demand, which would not exist in a fixed-diet environment. Flies housed in a choice environment, for example, are allowed to interact with each macronutrient individually, and this behavior would be expected to result in the sequential activation of sugar and protein sensing pathways. Flies exposed to a complete, fixed diet would only experience nutrients simultaneously. The perceived flavor of each macronutrient may be different when yeast and sugar are consumed separately compared to when they are paired together on a fixed diet (*Ahn et al., 2011*) and that may also influence lifespan and metabolism (*Ostojic et al., 2014*). Notably, when given a choice, flies consumed more sugar than yeast but spent the majority of time physically on the yeast food. Loss of 5-HT2A did not alter these behaviors, establishing that control flies are capable of recognizing the components of a choice diet and that animals lacking 5-HT2A retain this ability.

Perhaps the act of decision-making per se is costly? In nature, decisions are made continuously following evaluation of environmental resources to allow animals to adjust their behavior in a way that would maximize their fitness (*Galef, 1996*; *Lima, 1998*). In humans, an abundance of choices can be stressful, which is thought to reflect the perceived risk of making a bad choice and therefore losing out on other, more valuable resources (*Schwartz, 2004*; *Inbar et al., 2011*). Although acute stress can be beneficial (*Dhabhar et al., 2012*), chronic stress is often detrimental to lifespan and health (*Epel et al., 2004*; *Glaser and Kiecolt-Glaser, 2005*; *Lupien et al., 2009*). *Drosophila* exhibit stress-related behaviors that are mediated by neuronal signaling pathways that are conserved in mice and humans, including serotonin signaling (*Yang et al., 2013*; *Sachs et al., 2015*; *Mohammad et al., 2016*; *Ries et al., 2017*). It seems reasonable, therefore, to consider the possibility that 5-HT2A influences metabolic decisions in response to the perception or consumption of individual nutrients in ways that are important for aging. Biogenic amines, including serotonin and dopamine, have been shown to encode protein hunger and reward (*Ribeiro and Dickson, 2010*; *Vargas et al., 2010*; *Ro et al., 2016*; *Liu et al., 2017*), and serotonin receptor 5-HT2A has been implicated in mammalian stress (*Weisstaub et al., 2006*; *Sachs et al., 2015*) and in a putatively stressful situation in flies in which prolonged sight of dead conspecifics increases mortality and shortens lifespan (*Chakraborty et al., 2019*). Notably, a recent study in mice reported that unknown characteristics of each meal modulate aging independent of dietary composition or calories (*Mitchell et al., 2019*), suggesting that how animals recognize, interact with, or interpret their food may be a mechanism of aging that is conserved across taxa.

# Materials and methods

## Key resources table

| Reagent type (species) or resource | Designation | Source or reference | Identifiers | Additional information |
|---|---|---|---|---|
| Gene (*Drosophila melanogaster*) | *5-HT2A* | FlyBase | FLYB:FBgn0087012 | |
| Gene (*Drosophila melanogaster*) | *GDH* | FlyBase | FLYB:FBgn0001098 | |
| Genetic reagent (*Drosophila melanogaster*) | *w$^{1118}$* | Bloomington *Drosophila* Stock Center | | |
| Genetic reagent (*Drosophila melanogaster*) | *Canton-S* | Bloomington *Drosophila* stock Center | | |
| Genetic reagent (*Drosophila melanogaster*) | *5-HT2A$^{PL00052}$* | Bloomington *Drosophila* Stock Center | BDSC Cat# 19367, RRID:BDSC_19367 | Also see Materials and methods, section 1 |
| Genetic reagent (*Drosophila melanogaster*) | *w$^{Dahomey}$* | Other | | L. Partridge (University College London, UK) |
| Genetic reagent (*Drosophila melanogaster*) | *dFoxo$^{Δ94}$* | Other | BDSC Cat# 42220; RRID:BDSC_42220 | L. Partridge (University College London, UK) |
| Genetic reagent (*Drosophila melanogaster*) | *Tub-GS-GAL4* | Other | | R. Davis (The Scripps Research Institute, Jupiter, FL) |
| Genetic reagent (*Drosophila melanogaster*) | *elav-GAL4* | Bloomington *Drosophila* Stock Center | BDSC Cat# 458 | |
| Genetic reagent (*Drosophila melanogaster*) | *UAS-5-HT2A-RNAi$^{TRiP}$* | Bloomington *Drosophila* Stock Center | BDSC Cat# 31882; RRID:BDSC_31882 | |
| Genetic reagent (*Drosophila melanogaster*) | *UAS-5-HT2A-RNAi$^{KK}$* | Vienna *Drosophila* Resource Center | VDRC Cat# 102105; RRID:FlyBase_FBst0475582 | |
| Genetic reagent (*Drosophila melanogaster*) | *UAS-GDH-RNAi* | Bloomington *Drosophila* Stock Center | BDSC Cat# 51473; RRID:BDSC_51473 | |
| Genetic reagent (*Drosophila melanogaster*) | *UAS-eGFP* | Bloomington *Drosophila* Stock Center | BDSC Cat# 79025; RRID:BDSC_79025 | |
| Strain, strain background (*Drosophila melanogaster*) | 18In08-17 | Other | | P. Schmidt (University of Pennsylvania, PA) |
| Strain, strain background (*Drosophila melanogaster*) | 18Ln10-4 | Other | | P. Schmidt (University of Pennsylvania, PA) |
| Software, algorithm | DLife | Other | | S. Pletcher (University of Michigan) |
| Software, algorithm | DTrack | Other | | S. Pletcher (University of Michigan) |
| Software, algorithm | DDrop | Other | | S. Pletcher (University of Michigan) |
| Software, algorithm | RStudio | RStudio | | |

## Fly stocks

The standard laboratory stocks *w$^{1118}$*, *Canton-S*, and the mutant *5-HT2A$^{PL00052}$* were originally obtained from the Bloomington *Drosophila* Stock Center. As described in our previous publication (*Ro et al., 2016*), *5-HT2A$^{PL00052}$* were backcrossed 10 generations to *w$^{1118}$* prior to all the experiments. The *w$^{Dahomey}$* and *dFoxo$^{Δ94}$* lines were provided by L. Partridge (University College London, UK). *dFoxo$^{Δ94}$* were backcrossed into *w$^{Dahomey}$* prior to the lifespan experiments. Two isofemale lines (18In08-17 and 18Ln10-4, progeny from single wild females that were collected from two different orchards in Media, PA, in 2018) were gifts from P. Schmidt (University of Pennsylvania, PA). *Tub-GS-GAL4* was obtained from R. Davis (The Scripps Research Institute, Jupiter, FL). *elav-GAL4* (BDSC#458) and *UAS* transgenic lines, including *UAS-5-HT2A-RNAi$^{TRiP}$* (BDSC#31882), *UAS-5-HT2A-RNAi$^{KK}$* (VDRC#102105), *UAS-GDH-RNAi* (BDSC#51473), and *UAS-eGFP* (BDSC#79025), were purchased from either the Bloomington *Drosophila* Stock Center or the Vienna *Drosophila* Resource Center.

## Husbandry

All fly stocks were maintained on a standard cornmeal-based larval growth medium (produced by LabScientific Inc and purchased from Fisher Scientific) and in a controlled environment (25 °C, 60% humidity) with a 12:12 hr light:dark cycle. We controlled the developmental larval density by manually aliquoting 32 µl of collected eggs into individual bottles containing 25 ml of food. Following eclosion, mixed-sex flies were kept on SY10 (10% [w/v] sucrose and 10% [w/v] yeast) medium for 2–3 days until they were used for experiments. Pioneer table sugar (purchased from Gordon Food Service, MI) and MP Biomedicals Brewer's Yeast (purchased from Fisher Scientific) were used in our study.

## Dietary choice environment

To study the lifespan, behavior, and physiology of flies that have a dietary choice, we created food wells that were divided in the middle (*Figure 1A*). This allowed us to expose the flies to two separate sources of food simultaneously. For these experiments, we either loaded different foods on each side (choice diet) or the same food on both sides (fixed diet). Unless otherwise mentioned, the choice diets contained 10% (w/v) sucrose or 10% (w/v) yeast on either side, and the fixed diets contained a mix of 10% (w/v) sucrose and 10% (w/v) yeast on each side.

## Drug administration

We purchased all drugs from Sigma-Aldrich. In the pirenperone experiment, drug was initially dissolved in 100% dimethyl sulfoxide (DMSO) at 10 mM concentration, aliquoted and stored at −20°C. Prior to the experiment, 200 µM pirenperone and the same dilution of DMSO (control vehicle) were made. Then, 50 µl of the diluted drug or control vehicle was added to each food well. After the liquid evaporated (~2 hr), the food plates were ready for use as described in Survival assays.

In the GeneSwitch experiment, RU486 (mifepristone) was first dissolved in 80% (v/v) ethanol at 10 mM concentration, marked by blue dye (5% [w/v] FD and C Blue No. 1; Spectrum Chemical), and stored at −20°C. For experimental food, 200 µM RU486 or the same dilution of control vehicle (80% [v/v] ethanol) was made from the stock and added to the food. To ensure flies consumed the same amount of RU486 between the choice diet and the fixed diet, drug was only administered in the yeast portion of the choice diet, since the yeast intake on a choice diet is indistinguishable from the sugar–yeast intake on the fixed diet (*Figure 1E*). Lifespan assay was carried out as described below.

## Survival assays

Lifespans were measured using established protocols (*Linford et al., 2013*). Normally, six replicate vials (~150 experimental flies) were established for each treatment. Flies were transferred to fresh media every 2-3 days, at which time dead flies were removed and recorded using the DLife system developed in the Pletcher Laboratory (*Linford et al., 2013*). Flies were kept in constant temperature (25°C) and humidity (60%) conditions with a 12:12 hr light:dark cycle.

## Consumption–excretion (feeding assays)

We used the protocol described previously (*Shell et al., 2018*). Experimental flies were co-housed in vials (10 flies per vial, 8–10 replicates for each group) for 2–3 days following eclosion and then sorted into individual sex cohorts. After exposure to the choice or fixed diet for 12 days (food was changed every 2–3 days), flies were transferred to the choice or fixed diet with 1% (w/v) FD and C Blue No. 1 in either (1) only the S10 (10% [w/v] sucrose) food in the choice diet; (2) only the Y10 (10% [w/v] yeast) food in the choice diet; or (3) food in both wells in the fixed diet SY10 (10% [w/v] sucrose and 10% [w/v] yeast). Vials were discarded if one or more dead flies were observed after the 24 hr feeding period. Excreted dye (ExVial) was collected by addition of 3 ml of Milli-Q water of vials followed by vortexing. Concentration of the ExVial dye in water extracts was determined by absorbance at 630 nm, which was used to infer macronutrient consumption.

## Fly Liquid-Food Interaction Counter feeding assays

Flies were tested on the Fly Liquid-Food Interaction Counter (FLIC) system as previously described to monitor ethanol feeding (*Ro et al., 2014*) after exposure to a choice or fixed diet for 10 days. We

filled the liquid-food reservoir with 15% (v/v) ethanol + 5% (w/v) sucrose in 4 mg/l $MgCl_2$. We synchronized the feeding state prior to loading flies onto the FLIC system. Flies were removed from their food environment to kimwipes with 2 ml of water 2 hr before their natural breakfast time. At the peak of their normal breakfast meal (lights on), we flipped flies back onto their respective choice or fixed diets and let them feed for 2 hr. Following breakfast feeding, flies were briefly fasted for 3 hr on a wet kimwipe with 2 ml of water. We began the FLIC recording software before aspirating flies into the system and analyzed food or ethanol interactions for 6 hr. Flies were anesthetized briefly on ice and manually aspirated into the *Drosophila* feeding monitors (DFMs). Each DFM was loaded with flies from at least two treatment groups to reduce technical bias from the DFM signals. FLIC data were analyzed using custom R code, which is available at wikiflic.com. Default thresholds were used for analysis except for the following: minimum feeding threshold = 20, minimum events = 1, and tasting threshold = (10, 20). Flies that had zero feeding events over the testing interval were removed from the analysis.

## Video tracking and analyses

Experimental males were reared on SY10 food for 2–3 days shortly after eclosion. We aspirated individual flies into the two-choice food arenas (diameter = 31 mm, either a choice diet or a fixed diet, N = 12 for each group) and let them adapt to the environment for 1 day. Then, 6 hr video recording (10:00–16:00, at two frames per second with a resolution at 528 × 320) was taken and analyzed with DTrack, a video analysis package developed in the Pletcher Laboratory. Using the same speed threshold published before (*Corrales-Carvajal et al., 2016*), we classified the movements of flies into resting or sleeping (speed ≤ 0.2 mm/s), micromovement (0.2 mm/s < speed ≤2 mm/s), and walking (speed > 2 mm/s). For positional preference assay, we aspirated individual male flies into the two-choice food arenas (N = 24 for each genotype) after the designated exposure to the choice diet for 7 days. Three hour video recordings were taken (at two frames per second with a resolution at 528 × 320) and analyzed with DTrack. The fraction of time flies spent on each macronutrient was computed and used for statistical analyses.

## Climbing activity assay

After the designated exposure to the choice or fixed diet for 10 days (food was changed every 2–3 days), male flies were transferred to the negative geotaxis apparatus using brief $CO_2$ anesthesia, after which they were allowed to recover for 30 min. We used an automated process in which flies were dropped from 24'. After freefall, which effectively knocked the flies to the bottom of the chamber, a video camera was triggered and individual fly movements were tracked for 10 s using the DDrop software developed in our laboratory. From the tracking data, we were able to calculate, for each male, the total distance traveled up in 10 s.

## Fly weight, TAG, and protein assays

After the designated exposure to the choice or fixed diet for 2 or 3 weeks (food was changed every 2–3 days), experimental flies were quickly frozen, collected into groups of five, weighed, and then homogenized in 200 µl of cold phosphate-buffered saline containing 0.1% Triton X-100 (IBI Scientific) for 30 s at 30 Hz using a QIAGEN TissueLyser. For TAG quantification, the homogenate (20 µl) was added into 200 µl of Infinity Triglyceride Reagent (Thermo Electron Corp.) and incubated at 37°C for 10 min with constant agitation. TAG concentrations were determined by the absorbance at 520 nm and estimated by a known triglyceride standard. For protein measurement, 5 µl of fly homogenate was incubated with 200 µl of (1:50) 4% (w/v) cupric sulfate/bicinchoninic acid solution (Novagen) at room temperature for 30 min. Protein concentrations were estimated by the absorbance at 562 nm through the comparison with bovine serum albumin standards. Average weight, TAG, and protein values were based on at least six independent biological replicates (of five flies each) from multiple vials.

## Starvation assay

After the designated exposure to the choice or fixed diet for 20 days (food was changed every 2–3 days), flies were transferred to fresh vials containing 1% (w/v) agar. The number of dead flies was recorded approximately every 2–5 hr using DLife system.

## Smurf assay

We investigated intestinal integrity using the Smurf assay (*Regan et al., 2016*; *Rera et al., 2012*). Experimental flies were grouped in vials (25 flies per vial and 12 replicates per each diet) 2–3 days following eclosion. Following exposure to a choice or fixed diet for 12 days (food was changed every 2–3 days and survivorship of $w^{1118}$ males on choice diet was roughly 70%), flies were transferred to S30 (30% [w/v] sucrose) food with 2.5% (w/v) blue dye (FD and C Blue No. 1; Spectrum Chemical) for 24 hr. A fly was counted as a Smurf when blue was observed outside of the digestive tract (*Regan et al., 2016*; *Rera et al., 2012*). The proportion of Smurf flies in each replicate was calculated and used as a single observation for statistical analysis.

## Egg laying assays

Flies were raised according to our standard husbandry protocols (see above), and following eclosion, males and females were allowed to mate on SY10 food for 48–72 hr. Eight females and eight males were then sorted into individual vials, which contained either a choice or a fixed diet. Egg counts were taken on days 1, 2, 5, and 8 after sorting. Flies were transferred to fresh SY food 24 hr prior to the counting.

## Western blot

After exposure to the choice or fixed diet for 20 days, experimental flies were flash frozen in liquid nitrogen. Heads and bodies were separated using a metal sieve on dry ice, and 10 heads were pooled for each biological replicate. Heads were first pulverized to a fine powder using a plastic pestle on dry ice. Protein extraction was carried out on ice using RIPA buffer (Sigma–Aldrich) supplemented with protease inhibitor cocktail (Sigma), phosphatase inhibitor cocktail (Sigma), sodium orthovanadate (NEB, 1 mM), and sodium fluoride (NEB, 1 mM). Ice cold buffer (105μl) was added to heads followed by immediate homogenization with a motorized pestle for 10 s on ice. Lysates were incubated on ice for 10 min followed by 15 s of sonication and centrifugation at 16,000 × g at 4°C for 10 min. Supernatants were recovered, and protein concentration was determined using the Pierce BCA Protein Assay Kit (Fisher Scientific). Thirty micrograms of protein lysate was added to 2× protein sample buffer (1 mM Tris–HCL pH 6.8, 10% sodium dodecyl sulphate [SDS], 1% bromophenol blue, and 1M dithiothreitol) and denatured at 95°C for 10 min. Protein was separated by SDS–polyacrylamide gel electrophoresis on a 4–12% gel (Bio-Rad) at 200 V for 30 min, followed by electrophoretic transfer to PVDF membrane at 70 V for 1 hr. Blots were incubated in 5% milk in 1% TBS-T at room temperature for 1 hr, followed by overnight incubation with primary antibodies overnight at 4°C (anti-dS6K gift from Thomas Neufeld; dilution at 1:3500, anti-pS6KThr398 Cell Signaling Technologies, #9209; dilution at 1:1000, anti-GAPDH; dilution 1:6000). Membranes were washed with 1% TBS-T and incubated with HRP-conjugated secondary antibodies (Abcam) at room temperature for 4 hr. Membranes were washed again with 1% TBS-T and then incubated briefly in ECL substrate (SuperSignal West Femto, ThermoFisher) before imaging.

## Metabolomics analysis

Male flies were exposed to the choice or fixed diet for 20 days (food was changed every 2–3 days) and then switched to the other diet (experimental groups) or the same diet (control groups). Heads were removed via vortexing and manually separated from the bodies. Heads or bodies were then homogenized for 20 s in 200 μl of a 1:4 (v:v) water:MeOH solvent mixture using the Fast Prep 24 (MP Biomedicals). Following the addition of 800 μl of methanol, the samples were incubated for 30 min on dry ice, then homogenized again. The mixture was spun at 13,000 RPM for 5 min at 4°C, and the soluble extract was collected into vials. This extract was then dried in a speedvac at 30°C for approximately 3 hr. Using a liquid chromatography triple quadrupole tandem mass spectrometry (LC–QQQ–MS) machine in the multiple reaction monitoring (MRM) mode, we targeted 205 metabolites in 25 important metabolic pathways, in both positive MS and negative MS modes.

After removing any metabolites missing from more than 8 of 76 head samples (10.5%) and 8 of 78 body samples (10.3%), we were left with 103 head metabolites and 125 body metabolites. Metabolite abundance for remaining missing values in this data set were log-transformed and imputed using the *k*-nearest neighbor algorithm with the impute package of R Bioconductor (http://

www.bioconductor.org). We then normalized the data to the standard normal distribution ($\mu = 0$, $\sigma^2 = 1$). PCA was performed using the made4 package of R Bioconductor.

Our PCA results suggest the choice diet can elicit two processes: (1) a 5-HT2A-dependent, fast-responding metabolic network that is associated with aging despite only accounting for a small number of variation in the metabolomes (2.0–2.2%) and (2) a 5-HT2A-independent, stable metabolic component that accounts for a significant amount of metabolome variation (10.2–11.2%). To distinguish these two groups of metabolites, we used a linear model to analyze the variance attributed to diet, genotype, and interaction of the two for each metabolite:

Metabolite abundance (Y) = Diet + Genotype + Diet * Genotype.

We found 18 of 103 total metabolites in the head and 19 of 125 total metabolites in the body are modulated by choice diet in a 5-HT2A-dependent manner (*Figure 4A,B*, ANOVA, $p_{Diet*Genotype} \leq 0.05$). To identify 5-HT2A independent change, we searched for metabolites that have a diet effect ($p_{Diet} \leq 0.05$) but did not show a diet–genotype interaction ($p_{Diet*Genotype} > 0.05$). In doing so, we found 45 and 46 metabolites in the head and body, respectively. We plotted a heatmap to illustrate the expression changes of the metabolite upon nutrient presentation using the R package gplots. The metabolites were mapped to biochemical pathways using MetaboAnalystR 3.0 (*Pang et al., 2020*). Because the small number of detectable compounds in our targeted metabolomics panel (N = 103 in heads or 125 in bodies) limits the power of the enrichment analysis, candidate pathways were therefore identified as those with the number of hits greater than 3.

## Statistics

Unless otherwise indicated, pairwise comparisons between different treatment survivorship curves (both lifespan and starvation resistance) were carried out using the statistical package R within DLife (*Linford et al., 2013*). p-Values were obtained using log-rank test. For testing the interaction between genotypes and diets, we used Cox-regression analysis to report p-value for the interaction term. Mortality rates were calculated using standard methods (*Partridge et al., 2005*). To test the effects of diet and genotype involved in behavior classification and metabolites, we performed two-way ANOVA (on the square root- or log-transformed values) followed by post hoc significance test. To test for a diet effect on gut integrity, climbing ability, weight, TAG/protein levels, nutrient assumption, and positional preference, we used Mann–Whitney U test. Two-tailed p-values were reported unless otherwise mentioned. We designed the layout of individual samples for lifespan, behavior, physiology, and metabolomic analyses, so non-biological factors (e.g. the location of samples) do not contribute to the differences between groups.

## Data availability

Source data for all quantifications shown in Data *Figures 1–5*, figures supplements, and the supplementary files are provided with the paper. Metabolomic raw data, analyses, and statistics can be obtained from *Supplementary files 3–4* and our GitHub repository (github.com/ylyu-fly/Metabolomics-FlyChoiceDiet; *Lyu, 2021*; copy archived at swh:1:rev:86daef110d9a1eb27262bc5310b54cc2d315f7c1).

## Acknowledgements

We thank the editors and reviewers for insightful comments that have resulted in a considerably improved manuscript. We would like to acknowledge the members of the Pletcher laboratory for their comments on the experimental designs and analyses, members of the Promislow laboratory for their valuable opinions on the manuscript draft, the fly kitchen and undergraduate research assistants Kyrinna Wei and Justin Eagy of the Pletcher laboratory for experimental food preparation and fly husbandry, and Drs. Cheng Peng and Tong Shen for suggestions on the interpretation of the metabolomics results. This research was supported by the US National Institutes of Health, National Institute on Aging (R01 AG051649, R01 AG030593 to SDP; R01 AG063371 to SDP and DELP; and R01 AG049494 and P30 AG013280 to DELP), the Glenn Medical Foundation (to SDP), the Burroughs Wellcome Fund Collaborative Research Travel Grant (No. BWF1017452 to YL), and National Science Foundation Graduate Research Fellowship Program (No. DGE 1256260 to KJW).

## Additional information

### Funding

| Funder | Grant reference number | Author |
|---|---|---|
| Burroughs Wellcome Fund | Collaborative Research Travel Grant BWF1017452 | Yang Lyu |
| National Science Foundation | Graduate Research Fellowship Program DGE 1256260 | Kristina J Weaver |
| National Institutes of Health | R01 AG049494 | Daniel EL Promislow |
| National Institutes of Health | R01 AG051649 | Scott D Pletcher |
| National Institutes of Health | P30 AG013280 | Daniel EL Promislow |
| National Institutes of Health | R01 AG030593 | Scott D Pletcher |
| National Institutes of Health | R01 AG063371 | Scott D Pletcher |
| Glenn Foundation for Medical Research | Paul F. Glenn Center for Aging Research | Scott D Pletcher |

The funders had no role in study design, data collection and interpretation, or the decision to submit the work for publication.

### Author contributions

Yang Lyu, Scott D Pletcher, Conceptualization, Investigation, Supervision, Formal analysis, Resources, Data curation, Software, Methodology, Project administration, Writing-original draft, Writing - review and editing, Funding acquisition; Kristina J Weaver, Formal analysis, Investigation, Visualization, Writing - original draft, Funding acquisition; Humza A Shaukat, Marta L Plumoff, Investigation; Maria Tjilos, Investigation, Writing - review and editing; Daniel EL Promislow, Resources, Software, Supervision, Funding acquisition, Methodology, Project administration, Writing - review and editing

### Author ORCIDs

Yang Lyu https://orcid.org/0000-0002-4618-9043
Scott D Pletcher https://orcid.org/0000-0002-4812-3785

### Decision letter and Author response

Decision letter https://doi.org/10.7554/eLife.59399.sa1
Author response https://doi.org/10.7554/eLife.59399.sa2

## Additional files

### Supplementary files

• Supplementary file 1. Summary of dietary choice effects on lifespan of flies from different wild-type strains. Values represent mean lifespan (SEM) in days for each cohort. p-Values are from log-rank ratio test.

• Supplementary file 2. Summary of 5-HT2A gene expression in various tissues. RNA-seq data are obtained from FlyAtlas2 (flyatlas.gla.ac.uk/FlyAtlas2/).

• Supplementary file 3. Statistics of metabolomic analysis.

• Supplementary file 4. Raw data, sample information, and quality controls of metabolomic data.

• Transparent reporting form

### Data availability

Source data for all quantifications shown in Data Figures 1-5, figures supplements and the supplementary files are provided with the paper. Metabolomic raw data, analyses and statistics can be obtained from Supplementary Files 3-4 and our GitHub repository (http://github.com/ylyu-fly/

Metabolomics-FlyChoiceDiet). Copy archived at https://archive.softwareheritage.org/swh:1:rev:86daef110d9a1eb27262bc5310b54cc2d315f7c1/.

The following dataset was generated:

| Author(s) | Year | Dataset title | Dataset URL | Database and Identifier |
|---|---|---|---|---|
| Lyu Y, Promislow DEL, Pletcher SD | 2016 | Head and body metabolomes in response to nutrient choice | https://github.com/ylyu-fly/Metabolomics-Fly-ChoiceDiet | Supplementary data, FlyChoiceDiet |

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
