## [Decision Letter]

**Acceptance summary:**

Your work highlights the importance of choice making in the regulation of organism lifespan and reveals the crucial role of serotonin signaling in this regulation. We believe that this finding will provide a new angle for us to understand how diet modulates aging and longevity. In addition, in order to provide a clear indication of the biological system under investigation, we would like to suggest a revision of the title, "*Drosophila* Serotonin 2A receptor signaling coordinates central metabolic processes to modulate aging in response to nutrient choice".

**Decision letter after peer review:**

Thank you for submitting your article "Serotonin 2A receptor signaling coordinates central metabolic processes to modulate aging in response to nutrient choice" for consideration by *eLife*. Your article has been reviewed by three peer reviewers, and the evaluation has been overseen by a Reviewing Editor and Jessica Tyler as the Senior Editor. The following individual involved in review of your submission has agreed to reveal their identity: David Walker (Reviewer #2).

The reviewers have discussed the reviews with one another and the Reviewing Editor has drafted this decision to help you prepare a revised submission.

Summary:

Lyu et al. present interesting findings that nutritional choice affects organism physiology and lifespan. They provide compelling evidence to support that such effects are not a simple result of altered nutrient consumed. They further implicated the serotonin receptor 5-HT2A in this regulation by the dietary choice. Through metabolomic analysis, they also revealed changes in the TCA cycle and amino acid metabolism. Although reviewers agree the concept of nutritional choice regulating longevity is novel and will be of great interest to the field of aging, they suggested additional data and edits would help strengthen this claim.

Essential revisions:

1) The key claim that the observed changes are due to nutritional choice requires additional experiments.

a) When macronutrients are provided separately, there might be difference in consumption and absorption, contributing to the observed phenotypes. The authors have assayed consumption, showing flies consuming more sugar and the same amount of yeast on the choice diet. But the change in absorption has not been ruled out. The authors should determine whether consumption and/or absorption of each nutrient is altered in the choice diet when compared to mixed diet and perform this comparison in young and middle-aged files. For example, assessment of fecundity and mTOR signaling will help find possible changes in protein intake.

b) The olfactory perception of yeast can be different between the choice diet and the mixed diet. The previous finding from the lab has revealed the importance of yeast smell on fly lifespan. To rule out the smell of yeast as a confounding factor, the olfactory mutants that are smell blind should be subject to the choice diet.

2) The mechanistic follow-up of metabolomic analysis is weak. To strengthen this area, the authors should examine whether protein absorption is altered (also mentioned in the point 1) and whether *GDH* RNAi flies eat more under the choice diet.

3) Linking 5HT2A receptor to the regulation by the choice diet has added mechanistic insight, which needs to be strengthened by provide sufficient rationale for its selection. Are there other receptors screened? Did any additional receptors exhibit potential interactions?

4) There are many changes in the metabolomic data. What is the rationale for focusing on the TCA cycle intermediates? In addition, all metabolomic data (with statistics) should be shown without cherry-picking.

5) Statistics should be provided to Figure 2F and G.

6) Method description should be revised, including nutrient consumption (Figure 2D), Smurf assay for intestinal integrity (adding the missing reference), behavioral analysis (Figure 2F and G).

7) Additional discussion should be included to address sex specificity and genetic background difference.

Reviewer #1:

Lyu et al., present a study in which the main claim is that the choice of different nutrients itself modulates behavior, metabolism and lifespan, rather than AMOUNT of nutrients themselves. The implication of this is that caloric restriction is only one means to modulate metabolism and lifespan; food choice is another. As such, this is an interesting idea. However, several major concerns (see points below) made it unclear whether the observed effects were a true reflection of choice, or of increased sugar consumption. Further, choice of 5HT2A receptor for further study was not seen as a substantial advance, given what is already known. Finally, it was unclear from the data analysis/presentation of the metabolomic data whether TCA cycle changes were the true driver of the observed effects.

1) In Figure 1, do they have a measure of how much time the flies spend on each nutrient source in the choice diet? Without this, how do you know the flies are actually making a choice? The whole phenotype/effect could be explained by flies dwelling mostly on the sugary food (which they prefer), which the authors have shown to reduce lifespan.

2) Although the authors show in Figure 1 that intestinal integrity and climbing ability are not compromised in a 2-week period on the choice diet, it would be important to show that these measures are not more compromised than the fixed diet in older flies that have been on the choice diet for longer: 10, 20 and 30 days to assess the true effect of CD-based reduced lifespan.

3) Are the differences in Figure 2F and G significant? No statistics are provided and the data do not look convincing; correspondingly the conclusion that the diet-dependent changes in locomotion are not causal to changes in lifespan does not seem warranted.

4) The rationale for choosing 5HT2A mutants is presented as having reduced protein preference in starved female flies, and increased lifespan under dietary choice. However, it is possible that the effects they observe in the present study are an aspect of the same observation (see point 1).

5) In Figure 3, it is important to show all the PCA data in graphic format, not just those that differed amongst the various conditions.

6) Many metabolites were altered in the different conditions (Figure 4). What is the rationale for choosing to focus on the TCA cycle intermediates? It appears that there are broader changes to amino acid metabolism. Also, fold changes are not mentioned and cutoffs not given.

Reviewer #2:

This is an interesting extension of previous work from the Pletcher lab, published in *eLife* (Ro et al., 2016). The concept of nutritional choice impacting organismal physiology and longevity is very interesting.

The challenge, of course, is to provide irrefutable evidence that it is the “choice” that causes the observed physiological changes. On the whole, the authors have done an admirable job to exclude potential confounding factors. I have, however, some remaining concerns/suggestions for alternative explanations.

I am happy to leave it to the authors as to whether to attempt to address these concerns with additional experiments and/or to discuss in the manuscript.

Figure 1:

The entire premise of the paper assumes that the observed effects are “largely independent of nutrients consumed”. Technically, the authors have done a very good job, when focusing on nutrient consumption.

However, there could be more to this than just food consumption.

Is it possible/likely that on the choice diet the flies are absorbing more yeast/protein? Isn't the Figure 1H data consistent with this?

i.e. When the macronutrients are provided separately, are there are differences in absorption?

In the fixed diet, sugar and yeast are always consumed together.

Is consuming yeast without sugar, even for short periods, detrimental to longevity because of “elevated protein uptake”?

My (limited) understanding of human nutrition is that it really matters which ingredients are paired together in terms of health benefits, e.g. pairing tomatoes with olive oil.

Another potential issue, that given the lab's history is particularly relevant, is the issue of olfaction.

Could the smell of yeast be more pronounced when provided separately? If so, could this be a confounding factor?

Two suggestions to investigate this hypothesis:

1) Assay fecundity, which is linked to protein intake in flies in choice vs. fixed diet

2) Assay TOR signaling which is also linked to protein intake

3) Examine olfactory mutants (smell blind)

If the authors could demonstrate no difference in these parameters, it would strengthen the claims of the paper.

Figure 2:

Figure 2A-E is *w^1118^* really an appropriate control? Precise *P* element revertant?

Not surprisingly, different genotypes appear to have varying degrees of reduced longevity in response to the “choice food”. I.e. *Canton S* in Figure 1, *w^1118^* in Figure 2 and the RNAi strain in Figure 5. In this case (Figure 5), the effects of the choice diet appear to be modest.

Can the authors speculate as to why the *5-HT2A* mutants are short-lived on the fixed diet?

The metabolic data presented (Figure 4 and 5) are interesting, but, there is little in terms of mechanistic understanding provided.

It is interesting that amino acid metabolism is upregulated in response to the choice diet. Again, as outlined above, couldn't this reflect elevated protein intake?

The *GDH* RNAi experiment is very interesting, but, there appears to be little follow up and/or validation. Some suggestions to validate and/or strengthen the claims:

– Validate that the RNAi line works “as advertised”.

– Do *GDH* RNAi flies also eat more under the choice diet? This would strengthen the claims.

– Does neuronal RNAi also alter the response?

Reviewer #3:

This is an important contribution to the field of aging, nutrition and neuroscience. It will be of great interest to the general scientific audience of *eLife* and I am highly supportive of its acceptance for publication.

In this manuscript, Lyu et al. report that it is how nutrients are presented, or the way that they are eaten, which modulates aging and physiology in the fruit fly, *Drosophila melanogaster*. The authors provide compelling evidence to demonstrate that such effects are independent of the amount of nutrients consumed and that neuronal expression of the serotonin receptor *5-HT2A* is required for metabolic changes induced by dietary choice, which include upregulation of the TCA cycle and amino acid metabolism. Based on global metabolic data they identify and show that one key enzyme in these processes (Glutamate Dehydrogenase) is causal in the effect of diet choice in lifespan.

The results are intriguing and, together with previously presented data related to the role of 5-HT2A for evaluating protein demand, these discoveries demonstrate that nutrient assessment per se, and/or the feeding choices that result, have significant biological impact on aging. The effects of dietary calories and composition on aging are well-known and highly studied, and it seems reasonable to consider that specific characteristics of each meal, such as the ways in which they are presented (perhaps involving sensory perception) or the pattern in which nutrients are consumed, are important as well. These mechanisms may also be relevant to understanding the effects of different types of fasting/feeding regimes that have experienced a resurgence of interest in research (e.g., Mitchell et al., 2019) and in the general public.

The data presented in Lyu et al. are extensive, and the authors do an outstanding job of disentangling the confounding effects of behavior, feeding, and lifespan, from dietary presentation. Most importantly, the authors have provide convincing evidence that changes in lifespan in flies given a choice diet are not caused solely by a change in nutrient intake (i.e., an increase in sugar consumption). They make the extraordinary observation that at least for flies and likely many other organisms, there something of great interest occurring in making choices in the environment that is outside of the standard foci of calories or amount consumed. The support for 5-HT2A receptor in this process is absolutely solid, as is the link to specific metabolic changes through metabolic analysis and genetic manipulation. These are all major strengths of the manuscript. The authors speculate that the psychological effect of choice or the act of decision making itself may somehow be costly, and it is these assertions that while perhaps less-well supported, nonetheless follow from their work and are well appropriate for speculation given their strong data.

---

## [Author Response]

Essential revisions:1) The key claim that the observed changes are due to nutritional choice requires additional experiments.a) When macronutrients are provided separately, there might be difference in consumption and absorption, contributing to the observed phenotypes. The authors have assayed consumption, showing flies consuming more sugar and the same amount of yeast on the choice diet. But the change in absorption has not been ruled out. The authors should determine whether consumption and/or absorption of each nutrient is altered in the choice diet when compared to mixed diet and perform this comparison in young and middle-aged files. For example, assessment of fecundity and mTOR signaling will help find possible changes in protein intake.

We appreciate that reviewers bring up this idea that the process of nutrient integration is complex, it happens at multiple levels and can be regulated by the characteristics of a diet. Here, and in the revised manuscript, we hope to assuage the concern about nutrient absorption by adding the fecundity and TOR data per reviewer’s requests. Indeed, neither these two measures indicate effects of dietary choice. In addition, we highlight the weight and total protein data (presented in Figure 1—figure supplement 3) to discuss the involvement of protein intake. Together, these data support the interpretation that differences in protein intake/absorption is not the cause of lifespan effects of a choice diet.

“Although protein intake was not different in flies fed different diets, it seemed possible that dietary choice improved protein absorption, which might be important considering the influential role of this macronutrient in reducing lifespan (Fontana and Partridge, 2015). […] Furthermore, molecular and physiological markers that are strongly affected by protein availability, including mTOR activity (Fontana and Partridge, 2015) and female fecundity (Jensen et al., 2015) were not altered by dietary choice (Figure 1—figure supplement 3).”

The process and effects of sugar absorption are, however, less clear than that of protein. While of interest, we believe that they are beyond the current scope of the manuscript considering the evidence that lifespan is distinct from molecular and physiological mechanisms of sugar effects, whether they be due to differences in sugar consumption or absorption.

b) The olfactory perception of yeast can be different between the choice diet and the mixed diet. The previous finding from the lab has revealed the importance of yeast smell on fly lifespan. To rule out the smell of yeast as a confounding factor, the olfactory mutants that are smell blind should be subject to the choice diet.

Our current, ongoing experiment establishes that olfactory-deficient mutants exhibit pronounced lifespan effects of a choice effects, excluding the involvement of yeast smell as a confounding factor. We will be happy to provide the data in the final version if the editors and reviewers feel this is an essential component of this study.

**Author response image 1. sa2fig1:** 

2) The mechanistic follow-up of metabolomic analysis is weak. To strengthen this area, the authors should examine whether protein absorption is altered (also mentioned in the point 1) and whether *GDH* RNAi flies eat more under the choice diet.

In addition to new information about protein absorption (see above point 1a), we have provided new data on the effects of *GDH* knockdown. Neither feeding nor total protein in the fly was affected by *GDH* knock-down. We revised the text as follows:

“*GDH* knockdown had no effect on nutrient consumption (Figure 5D) nor total protein abundance in the fly (Figure 5—figure supplement 2). Which tissues/cells are critical for the diet-specific lifespan effects that induced by *GDH* knockdown, and whether such influences are due to changes in α-ketoglutarate per se or to alterations in the metabolic network (Hoffman et al., 2017) more broadly, remain to be determined.”

3) Linking 5HT2A receptor to the regulation by the choice diet has added mechanistic insight, which needs to be strengthened by provide sufficient rationale for its selection. Are there other receptors screened? Did any additional receptors exhibit potential interactions?

Our choice of the 5-HT2A receptor was based on earlier studies that identified this receptor as a modulator of protein reward and originally documented diet-specific effects on lifespan (Ro et al., 2016). Our goal in the manuscript was to expand our understanding of the behavioral, physiological and aging effects of the choice diet, to determine the extent to which this effect was present in males, and to study the underlying mechanisms at play in this pathway. Loss of function in other serotonin receptors had no influence on protein preference (Ro et al., 2016), however it remains to be tested whether they might influence the aging processes in a choice environment through unknown mechanisms. We have revised text to the manuscript that clarifies this:

“Previously we hypothesized that neuronal mechanisms involved in nutrient evaluation might influence lifespan when macronutrients are presented separately (Ro et al., 2016). […] Here we focused on the elucidating the mechanisms underlying the effects of the 5-HT2A receptor and expanded our analysis of its effects on dietary choice and male lifespan.”

4) There are many changes in the metabolomic data. What is the rationale for focusing on the TCA cycle intermediates? In addition, all metabolomic data (with statistics) should be shown without cherry-picking.

We now include the statistics for all metabolites in the new Supplementary file 3. The rationale for investigating the TCA cycle has been added to the manuscript:

“Furthermore, MetaboAnalystR identified related pathways, involving the synthesis of aminoacyl-tRNAs or specific amino acids, which were found either in both tissues or only in the body (Figure 4D). […] A closer look into its metabolic map (KEGG ID: map00250) revealed that candidate metabolites are either members of the TCA cycle or are the direct precursors of TCA metabolites, which indicated the modulation of this metabolic process in response to dietary choice.”

5) Statistics should be provided to Figure 2F and G.

These statistics were provided in the original manuscript, but we realize the difficulty in identifying them. They are now highlighted in pink boxes.

6) Method description should be revised, including nutrient consumption (Figure 2D), Smurf assay for intestinal integrity (adding the missing reference), behavioral analysis (Figure 2F and G).

We have revised the Materials and methods accordingly for nutrient consumption, Smurf assay and for behavioral analysis, and the missing reference has been added now.

7) Additional discussion should be included to address sex specificity and genetic background difference.

We appreciated the general genetics context that the reviewer provided with us. A new paragraph that focuses on sexual dimorphism and genetic background has now been added in the Discussion:

“We found that dietary choice influenced lifespan in both male and female laboratory flies and that 5-HT2A is required for this effect in both sexes (see female data in Ro et al., 2016), suggesting the mechanisms are at least partially shared between them. […] Future studies that focus on dissecting the role of individual macronutrients in the choice effects or on investigating the involvement of sex-specific nutrient sensing proteins (see Hudry et al., 2019), perhaps with a focus on population genetics of natural populations, will be useful for understanding the nature of sexual dimorphism on lifespan and physiology.”

Reviewer #1:Lyu et al., present a study in which the main claim is that the choice of different nutrients itself modulates behavior, metabolism and lifespan, rather than AMOUNT of nutrients themselves. The implication of this is that caloric restriction is only one means to modulate metabolism and lifespan; food choice is another. As such, this is an interesting idea. However, several major concerns (see points below) made it unclear whether the observed effects were a true reflection of choice, or of increased sugar consumption. Further, choice of 5HT2A receptor for further study was not seen as a substantial advance, given what is already known. Finally, it was unclear from the data analysis/presentation of the metabolomic data whether TCA cycle changes were the true driver of the observed effects.1) In Figure 1, do they have a measure of how much time the flies spend on each nutrient source in the choice diet? Without this, how do you know the flies are actually making a choice? The whole phenotype/effect could be explained by flies dwelling mostly on the sugary food (which they prefer), which the authors have shown to reduce lifespan.

Our positional preference data were shown in Figure 2E, where we observed that flies (both controls and mutants) spend most of their time on the protein side. Furthermore, the positional preference is 5-HT2A-independent and therefore does not link mechanistically with the lifespan effects.

2) Although the authors show in Figure 1 that intestinal integrity and climbing ability are not compromised in a 2-week period on the choice diet, it would be important to show that these measures are not more compromised than the fixed diet in older flies that have been on the choice diet for longer: 10, 20 and 30 days to assess the true effect of CD-based reduced lifespan.

It is unclear to us what hypotheses the reviewer is seeking to distinguish. We specifically used these two assays (Figure 1C and D) to directly test whether the choice diet induces acute effects on general health. In our opinion, effects on long term health are more difficult to determine because inference of this sort would be directly confounded by known differences in the rates of aging between diets.

3) Are the differences in Figure 2F and G significant? No statistics are provided and the data do not look convincing; correspondingly the conclusion that the diet-dependent changes in locomotion are not causal to changes in lifespan does not seem warranted.

The relevant *P*-values from a two-way ANOVA were originally presented above each panel. Recognizing the complexity of this figure, however, we have now highlighted the values in pink boxes.

4) The rationale for choosing *5HT2A* mutants is presented as having reduced protein preference in starved female flies, and increased lifespan under dietary choice. However, it is possible that the effects they observe in the present study are an aspect of the same observation (see point 1).

Contrary to our previously published work in which flies were purposefully starved, we find no evidence that flies fed a choice diet are starved or nutrient (protein) deprived: they spend most of their time on the yeast food (see response to point #1 above); they exhibit normal levels of egg production and TOR signaling; and they are of normal weight. It is possible that mechanisms of nutrient perception or that cell-nonautonomous effects of perceived protein reward underlie the mechanisms through which lifespan is affected, and if so, that would be a significant advance. Similarly, we document 5-HT2A-dependent metabolic changes in response to dietary choice, among which αKG seems to be causal for effects on lifespan. Whether neuronal and perception effects are common to both phenomena remain to be determined, but either way, in our opinion the data presented in this manuscript highlight new mechanisms of aging.

5) In Figure 3, it is important to show all the PCA data in graphic format, not just those that differed amongst the various conditions.

The clustering patterns of major PC combinations (PC1-PC6, which account for 85% (in head) or 87% (in body) of the total variance), as well as the proportion of variance for each PC, have been added as Figure 3—figure supplement 1 (for head data) and Figure 3—figure supplement 2 (for body data).

6) Many metabolites were altered in the different conditions (Figure 4). What is the rationale for choosing to focus on the TCA cycle intermediates? It appears that there are broader changes to amino acid metabolism. Also, fold changes are not mentioned and cutoffs not given.

We have added detailed statistics for each metabolite in Supplementary file 3. The full datasets and analyses can also be accessed from GitHub (github.com/ylyu-fly/Metabolomics-FlyChoiceDiet). As noted in the main text, we used *P*-values, not fold-changes as cut-offs:

“Criteria for inclusion in Group A consisted of a statistically significant effect of diet (*P*_Diet_ < 0.05) and a non-significant diet x genotype interaction (*P*_Diet * Genotype_ > 0.05).”

“We identified 18 out of 103 total metabolites in the head and 19 out of 125 total metabolites in the body that were present in this group (Figure 4B, ANOVA, *P*_Diet * Genotype_ ≤ 0.05).”

We have also added text to the Results describing our rationale for focusing on TCA cycle.

Reviewer #2:This is an interesting extension of previous work from the Pletcher lab, published in eLife (Ro et al., 2016). The concept of nutritional choice impacting organismal physiology and longevity is very interesting.The challenge, of course, is to provide irrefutable evidence that it is the “choice” that causes the observed physiological changes. On the whole, the authors have done an admirable job to exclude potential confounding factors. I have, however, some remaining concerns/suggestions for alternative explanations.I am happy to leave it to the authors as to whether to attempt to address these concerns with additional experiments and/or to discuss in the manuscript.Figure 1:The entire premise of the paper assumes that the observed effects are “largely independent of nutrients consumed”. Technically, the authors have done a very good job, when focusing on nutrient consumption.However, there could be more to this than just food consumption.Is it possible/likely that on the choice diet the flies are absorbing more yeast/protein? Isn't the Figure 1H data consistent with this?i.e. When the macronutrients are provided separately, are there are differences in absorption?In the fixed diet, sugar and yeast are always consumed together.Is consuming yeast without sugar, even for short periods, detrimental to longevity because of “elevated protein uptake”?My (limited) understanding of human nutrition is that it really matters which ingredients are paired together in terms of health benefits, e.g. pairing tomatoes with olive oil.Another potential issue, that given the lab's history is particularly relevant, is the issue of olfaction.Could the smell of yeast be more pronounced when provided separately? If so, could this be a confounding factor?Two suggestions to investigate this hypothesis:1) Assay fecundity, which is linked to protein intake in flies in choice vs. fixed diet2) Assay TOR signaling which is also linked to protein intake3) Examine olfactory mutants (smell blind)If the authors could demonstrate no difference in these parameters, it would strengthen the claims of the paper.

We appreciate the insights and critiques provided by this reviewer, particularly those focused on protein sensing and absorption in the dietary choice effects, as we recognize that many uncharacterized features of diets may modulate aging. We have worked hard during difficult times in the lab to add data from new experiments that address these issues in the revised manuscript:

a) Specifically, three new experiments provide insight into the point about the possibility of diet-dependent protein absorption. Briefly, we measured fecundity and TOR activity (based on phosphorylation status of S6K) and neither measure indicated an effect of dietary choice. Furthermore, total protein in the fly was also not affected by protein choice (nor was weight), which also indicates that, given similar amounts of protein intake, protein absorption/utilization is not different between diets. The manuscript has been revised accordingly (Results).

b) This reviewer raised an interesting point that the choice effects could be the result of “pairing” different nutrients/foods. We find this (and other effects of patterning in feeding) to be very intriguing, and to our knowledge, it is largely unknown how flies adjust feeding behaviors in complex environments (but see Ahn et al., 2011). We believe that the choice diet paradigm (and modifications to it) provides an excellent opportunity to investigate such mechanisms in the future. For now, we are happy to include this possibility in the Discussion, as show below:

“Flies housed in a choice environment, for example, are allowed to interact with each macronutrient individually, and this behavior would be expected to result in the sequential activation of sugar and protein sensing pathways. […] The perceived flavor of each macronutrient may be different when yeast and sugar are consumed separately compared to when they are paired together on a fixed diet (Ahn et al., 2011), and that may also influence lifespan and metabolism (Ostojic et al., 2014).”

c) We agree that this would be a good experiment test the role of sensory perception in dietary choice effects. While the characterization of sensory requirements is complex and is well beyond the focus of a single manuscript (e.g., if you can’t smell or taste the food can you extract reward from it?), our current ongoing experiments clearly indicate that it is not simply a more odorant rich environment that is reducing lifespan upon dietary choice. This experiment is on-going, although the results are clear (Author response image 1). We will be willing to include the final data in a final version of the manuscript if the editors and reviewers feel it is an essential. Indeed, a more complete investigation of external and internal sensory mechanisms, together with patterned feeding, is a likely next step, although, we believe, outside the scope of the current manuscript.

Figure 2:Figure 2A-E is w^1118^ really an appropriate control? Precise P element revertant?

We appreciate and recognize the importance of controlling for background effects. We chose in this instance to use extensive backcrossing of *5-HT2A* mutants (ten generations; as described in the Materials and methods) together with background-independent RNAi. Precise excision (together with backcrossing) is also a reasonable approach, which we chose not to pursue. The primary reason for this is the potential for unintended genetic effects of excision and the time and effort required to achieve it. We felt the two manipulations described above provided strong inference.

Not surprisingly, different genotypes appear to have varying degrees of reduced longevity in response to the “choice food”. I.e. *Canton-S* in Figure 1, *w^1118^* in Figure 2 and the RNAi strain in Figure 5. In this case (Figure 5), the effects of the choice diet appear to be modest.

We acknowledged the observations on the effects of genetic background, where we added a new paragraph in the Discussion on this topic.

Can the authors speculate as to why the *5-HT2A* mutants are short-lived on the fixed diet?

A challenge here is that serotonin signaling has many pleotropic effects, although the extent to which some/all are 5-HT2A dependent is unclear. In *Drosophila* to name a few, 5-HT2A is involved in circadian rhythms (Nichols et al., 2006), walking (Howard et al., 2019), feeding (as shown in this paper, Gasque et al., 2013 and Ro et al., 2016), and sensory perception of dead conspecifics (Chakraborty et al., 2019). All of these effects could potentially contribute to its lifespan effect on a fixed diet. We have added some discussion of this to the revision.

“Mutant flies were modestly short-lived on the standard, fixed diet, which is likely due to the pleotropic effects of this receptor (Gasque et al., 2013; Chakraborty et al., 2019; Howard et al., 2019).”

The metabolic data presented (Figures 4 and 5) are interesting, but, there is little in terms of mechanistic understanding provided.It is interesting that amino acid metabolism is upregulated in response to the choice diet. Again, as outlined above, couldn't this reflect elevated protein intake?

As related to our responses to the point #1, there is no evidence showing that flies on a choice diet have elevated protein intake. The upregulation of amino acid metabolism therefore seems to reflect differences in the way that a fixed amount of amino acids are metabolized (also see our response to reviewer #1, point #6).

The *GDH* RNAi experiment is very interesting, but, there appears to be little follow up and/or validation. Some suggestions to validate and/or strengthen the claims:– Validate that the RNAi line works “as advertised”.

qPCR data reveal variable knockdown (up to 20% decrease) in *GDH* transcript amount, indicating that either partial knock-down is sufficient to influence lifespan or that the RNAi may function at post-transcriptional levels. We interpret this as implying that this particular pathway is subject to complex regulatory mechanisms (Smith et al., 2019). Indeed, our results simply provide one direction forward in identifying specific molecules that are responsible for the aging effect, which we believe is a significant advance. Given the difficulty of laboratory work, we would argue that detailed molecular analyses of multiple components of this pathway are beyond the scope of the current contribution.

– Do *GDH* RNAi flies also eat more under the choice diet? This would strengthen the claims.

We have provided new data on the food intake and protein absorption of *GDH* RNAi flies. In short, no, neither of these factors seems to contribute to the lifespan effects. Please see our revised text below:

“*GDH* knockdown had no effect on nutrient consumption (Figure 5D) nor total protein abundance in the fly (Figure 5—figure supplement 2).”

– Does neuronal RNAi also alter the response?

We are interested in testing the tissue requirement of *GDH* knockdown, but respectively argue that such experiments are best addressed in a follow-up manuscript (see our second response to point 3, above). Nevertheless, the relevance of this point is now addressed in the revision:

“Which tissues/cells are critical for the diet-specific lifespan effects that induced by *GDH* knockdown, and whether such influences are due to changes in α-ketoglutarate per se or to alterations in the metabolic network (Hoffman et al., 2017) more broadly, remain to be determined.”